# Neuronal complexity is attenuated in preclinical models of migraine and restored by HDAC6 inhibition

Zachariah Bertels[1], Harinder Singh[2], Isaac Dripps[1], Kendra Siegersma[1], Alycia F Tipton[1], Wiktor D Witkowski[1], Zoie Sheets[1], Pal Shah[1], Catherine Conway[1], Elizaveta Mangutov[1], Mei Ao[2], Valentina Petukhova[3], Bhargava Karumudi[3], Pavel A Petukhov[3], Serapio M Baca[4,5], Mark M Rasenick[1,2,6], Amynah A Pradhan[1]*

[1]Department of Psychiatry, University of Illinois at Chicago, Chicago, United States; [2]Department of Physiology and Biophysics, University of Illinois at Chicago, Chicago, United States; [3]Department of Medicinal Chemistry and Pharmacognosy, University of Illinois at Chicago, Chicago, United States; [4]Department of Pharmaceutical Sciences, University of Colorado Anschutz Medical Campus, Aurora, United States; [5]Department of Neurology, University of Colorado Anschutz Medical Campus, Aurora, United States; [6]Jesse Brown VAMC, Chicago, United States

**Abstract** Migraine is the sixth most prevalent disease worldwide but the mechanisms that underlie migraine chronicity are poorly understood. Cytoskeletal flexibility is fundamental to neuronal-plasticity and is dependent on dynamic microtubules. Histone-deacetylase-6 (HDAC6) decreases microtubule dynamics by deacetylating its primary substrate, $\alpha$-tubulin. We use validated mouse models of migraine to show that HDAC6-inhibition is a promising migraine treatment and reveal an undiscovered cytoarchitectural basis for migraine chronicity. The human migraine trigger, nitroglycerin, produced chronic migraine-associated pain and decreased neurite growth in headache-processing regions, which were reversed by HDAC6 inhibition. Cortical spreading depression (CSD), a physiological correlate of migraine aura, also decreased cortical neurite growth, while HDAC6-inhibitor restored neuronal complexity and decreased CSD. Importantly, a calcitonin gene-related peptide receptor antagonist also restored blunted neuronal complexity induced by nitroglycerin. Our results demonstrate that disruptions in neuronal cytoarchitecture are a feature of chronic migraine, and effective migraine therapies might include agents that restore microtubule/neuronal plasticity.

*For correspondence:
Pradhan4@uic.edu

Competing interests: The authors declare that no competing interests exist.

## Introduction

Migraine is an extremely common neurological disorder that is estimated to affect 14% of the global population, making it the third most prevalent disease worldwide (*Global Burden of Disease Study 2013 Collaborators, 2015*; *GBD 2016 Disease and Injury Incidence and Prevalence Collaborators, 2017*). One particularly debilitating subset of migraine patients are those with chronic migraine, which is defined as having more than 15 headache days a month (*Headache Classification Committee of the International Headache Society (IHS), 2013*). Despite its high prevalence, migraine therapies are often only partially effective or are poorly tolerated, creating a need for better pharmacotherapies (*Ford et al., 2017*). Recent clinical success of antibodies and antagonists against calcitonin gene-related peptide (CGRP) and CGRP receptor demonstrate the effectiveness of targeted migraine therapeutics. While there has been more research into understanding the molecular mechanisms of migraine, there remains much to be discovered.

**eLife digest** Migraines are a common brain disorder that affects 14% of the world's population. For many people the main symptom of a migraine is a painful headache, often on one side of the head. Other symptoms include increased sensitivity to light or sound, disturbed vision, and feeling sick. These sensory disturbances are called aura and they often occur before the headache begins. One particularly debilitating subset of migraines are chronic migraines, in which patients experience more than 15 headache days per month. Migraine therapies are often only partially effective or poorly tolerated, making it important to develop new drugs for this condition, but unfortunately, little is known about the molecular causes of migraines.

To bridge this gap, Bertels et al. used two different approaches to cause migraine-like symptoms in mice. One approach consisted on giving mice nitroglycerin, which dilates blood vessels, produces hypersensitivity to touch, and causes photophobia in both humans and mice. In the second approach, mice underwent surgery and potassium chloride was applied onto the dura, a thick membrane that surrounds the brain. This produces cortical spreading depression, an event that is linked to migraine auras and involves a wave of electric changes in brain cells that slowly propagates across the brain, silencing brain electrical activity for several minutes.

Using these approaches, Bertels et al. studied whether causing chronic migraine-like symptoms in mice is associated with changes in the structures of neurons, focusing on the effects of migraines on microtubules. Microtubules are cylindrical protein structures formed by the assembly of smaller protein units. In most cells, microtubules assemble and disassemble depending on what the cell needs. Neurons need stable microtubules to establish connections with other neurons. The experiments showed that provoking chronic migraines in mice led to a reduction in the numbers of connections between different neurons. Additionally, Bertels et al. found that inhibiting HDAC6 (a protein that destabilizes microtubules) reverses the structural changes in neurons caused by migraines and decreases migraine symptoms. The same effects are seen when a known migraine treatment strategy, known as CGRP receptor blockade, is applied.

These results suggest that chronic migraines may involve decreased neural complexity, and that the restoration of this complexity by HDAC6 inhibitors could be a potential therapeutic strategy for migraine.

Neuroplastic changes play an important role in a variety of chronic neuropsychiatric conditions (*Descalzi et al., 2015*), and epigenetic alterations through histone deacetylases (HDACs) are frequently investigated. HDACs are best characterized for their ability to deacetylate histones, promoting chromatin condensation and altered gene expression (*Krishnan et al., 2014*). Intriguingly, some HDACs can also deacetylate non-histone targets, including proteins involved in cytoarchitecture and dynamic cellular structure. Due to its cytoplasmic retention signal, HDAC6 is primarily expressed in the cytosol (*Valenzuela-Fernández et al., 2008*), and one of its primary targets for deacetylation is α-tubulin (*Valenzuela-Fernández et al., 2008*; *d'Ydewalle et al., 2012*). α- and β-tubulin form heterodimers that make up microtubules, which are a major component of the cytoskeleton, and regulate intracellular transport, cell morphology, motility, and organelle distribution (*Janke and Bulinski, 2011*; *Janke and Montagnac, 2017*). Microtubules undergo multiple cycles of polymerization and depolymerization, creating a state of dynamic instability (*Janke and Bulinski, 2011*). Tubulin displays a variety of post-translational modifications, including α-tubulin acetylation, which occurs endogenously through α-tubulin N-acetyltransferase I (αTAT1), and it is correspondingly deacetylated by HDAC6 (*Janke and Montagnac, 2017*). Tubulin acetylation is associated with increased flexibility and stability of microtubules (*Xu et al., 2017*). In contrast, deacetylated microtubules are more fragile and prone to breakage or increased cycling between assembly and disassembly (*Xu et al., 2017*). Microtubules are important for cellular response to injury and play a role in neurite branching (*Gallo, 2011*); and microtubule dynamics influence neuronal signaling and mediate axonal transport (*Covington et al., 2009*; *Braun et al., 2011*; *Schappi et al., 2014*; *Jin et al., 2017*; *Van Helleputte et al., 2018*). Importantly, changes in cellular structure such as alterations in dendritic spine density, have been implicated in disease chronicity (*Jochems et al., 2015*; *Forrest et al., 2018*; *Singh et al., 2018*; *Nestler and Lüscher, 2019*).

The aims of this study were to determine if altered neuronal cytoarchitecture facilitates the chronic migraine state and whether modifying this by inhibition of HDAC6 would be effective in mouse models of migraine. We observed decreased neuronal complexity in headache-processing brain regions in the nitroglycerin (NTG) model of chronic migraine-associated pain. We further demonstrated that treatment with HDAC6 inhibitor reversed these cytoarchitectural changes and correspondingly decreased cephalic allodynia. These studies were extended to a mechanistically distinct model of migraine, cortical spreading depression (CSD), which is thought to be the electrophysiological correlate of migraine aura. Again, we observed decreased neuronal complexity in migraine related sites, which was reversed by a HDAC6 inhibitor. To investigate the translational implication, we also tested the effect of olcegepant, a CGRP receptor inhibitor, and found that it alleviated chronic allodynia induced by NTG, and restored cytoarchitectural changes associated with chronic migraine-associated pain. These results suggest a novel mechanism for migraine pathophysiology and establish HDAC6 as a novel therapeutic target for this disorder.

## Results

### Exposure to chronic NTG induces cytoarchitectural changes in key pain processing regions

Changes in the structural plasticity of neurons have been observed in a number of neuropsychiatric disorders and can serve as a marker of disease chronicity (*Forrest et al., 2018*; *Nestler and Lüscher, 2019*). To investigate if this was also the case in migraine, we treated male and female C57BL/6J mice every other day for 9 days with NTG or vehicle and tested for periorbital mechanical responses on days 1, 5, and 9 (*Figure 1A*). NTG induced severe and sustained cephalic allodynia as measured by von Frey hair stimulation of the periorbital region as compared to vehicle animals on the same day (*Figure 1B*). Mice were sacrificed on day 10, 24 hr after the final NTG/VEH treatment, and neuronal size and arborization were examined through a Golgi staining procedure in a key cephalic pain processing region, the trigeminal nucleus caudalis (TNC) (*Figure 1C*; *Goadsby et al., 2017*). We observed a dramatic decrease in neuronal complexity after NTG treatment (*Figure 1D*). Neurons of chronic NTG-treated mice had significantly fewer branch points (*Figure 1E*), and shorter neurites resulting in decreased overall length of the neurons (*Figure 1F*). Further examination of the complexity of the neurons using Sholl Analysis, showed a significant decrease in the number of intersections following NTG treatment (*Figure 1G–I*). In addition to the TNC we also determined if other brain regions related to central pain processing were affected by chronic NTG treatment. We examined the somatosensory cortex (SCx) and periaqueductal gray (PAG) of these mice and found similar results, where neurons from NTG-treated mice had fewer branch points, were shorter in length, and had fewer intersections (*Figure 1—figure supplement 1A–D and E–H*, respectively). To ensure that this effect was associated with migraine-pain processing and not a non-specific effect of NTG we also analyzed neuronal complexity in the nucleus accumbens shell (NAc), a region more commonly associated with reward, and found no alteration in number of branches, total neuron length, or Sholl analysis for cells in this region (*Figure 1—figure supplement 1I–L*). Furthermore, we also examined the dorsal horn of the lumbar and cervical spinal cord, important sites for pain processing in the peripheral nervous system. No differences in NTG versus vehicle controls in number of branches, total neuron length, or Sholl analysis were observed (*Figure 1—figure supplement 1M–T*). These results suggest that decreased neuronal complexity may be a feature that maintains the chronic migraine state; a previously undiscovered phenomenon.

### HDAC6 inhibition increases acetylated α-tubulin and neuronal cytoarchitectural complexity

We hypothesized that if we could restore migraine-compromised cytoarchitectural complexity, this might also relieve cephalic pain. Recent studies demonstrate that increased tubulin acetylation facilitates microtubule flexibility and prevents microtubule breakage (*Janke and Montagnac, 2017*; *Portran et al., 2017*; *Xu et al., 2017*). Thus, we hypothesized that inhibiting HDAC6 to promote microtubule stability may restore the neuronal complexity observed following chronic NTG. We tested the selective HDAC6 inhibitor, ACY-738, in the chronic NTG model. Mice were treated with NTG or VEH for 9 days. On day 10, mice were injected with ACY-738, after 4 hr, tissue was

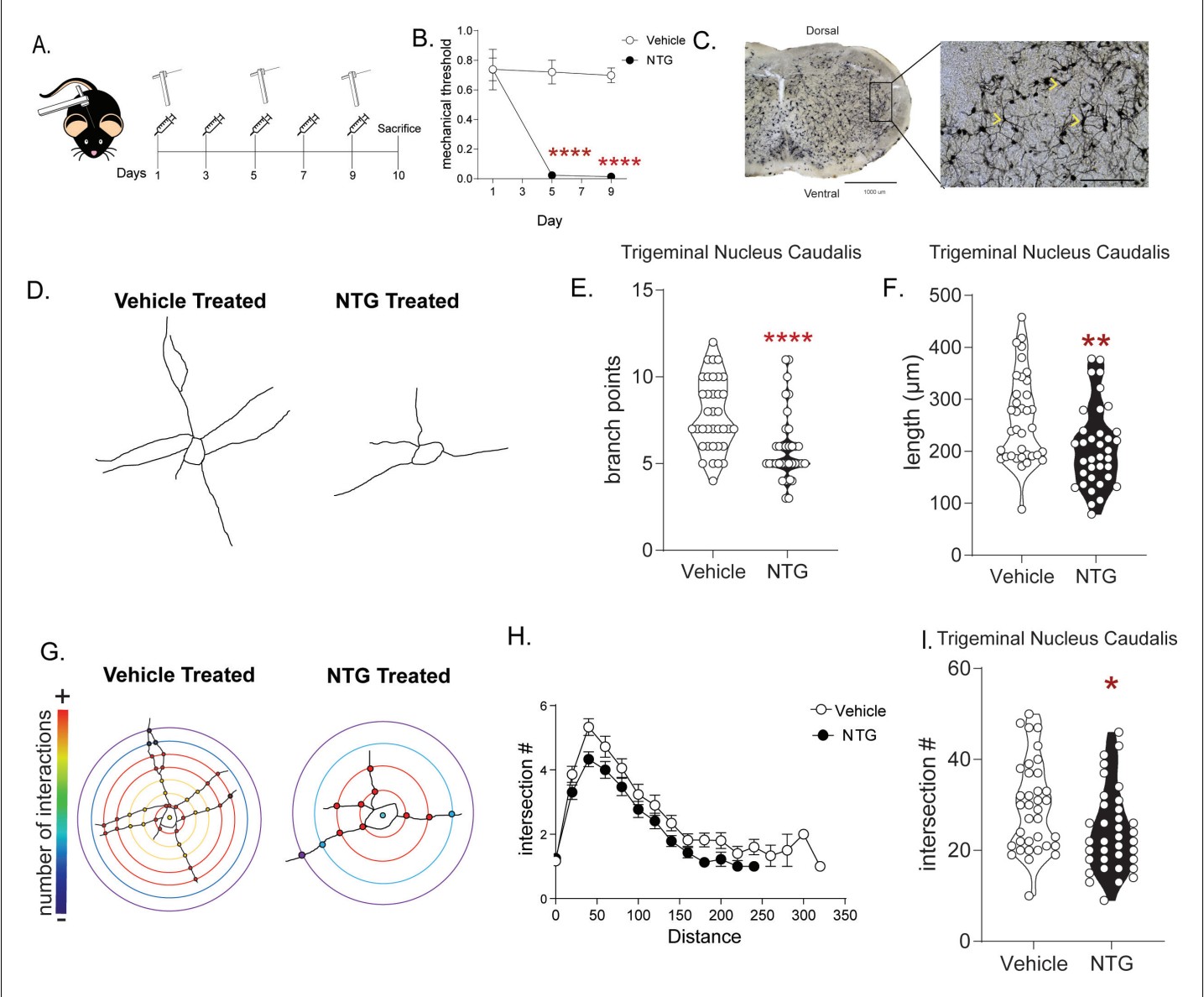

**Figure 1.** The NTG model of chronic migraine produces cytoarchitectural changes in a cephalic pain processing region. (A) Schematic of testing schedule, M and F C57Bl/6J mice were treated with vehicle or nitroglycerin (10 mg/kg, IP; NTG) every other day for 9 days. (B) Periorbital mechanical thresholds were accessed prior to Vehicle/NTG administration on days 1, 5, and 9. NTG produced severe cephalic allodynia p<0.0001 effect of drug F = 44.62, time F = 19.16, and interaction F = 16.28, two-way RM ANOVA and Holm-Sidak post hoc analysis. ****p<0.0001, relative to vehicle on day 1; n = 8/group. (C) Representative image taken of Golgi stained TNC at 4x (left) and 20x (right). Chevrons indicate neurons traced from the selected image and demonstrate type of neurons selected from this region. (D) Representative tracing of neurons from mice treated with chronic vehicle (left) or NTG (right) demonstrating reduced neural processes after chronic NTG treatment. (E) The number of branch points/neuron was decreased following chronic NTG. Unpaired t-test. ****p<0.0001, t = 4.029, df = 70, 95% CI (−2.866,–0.9678). (F) Total neuron length was also compromised after NTG treatment. Unpaired t-test. **p=0.0066, t = 2.798, df = 70, 95% CI (−93.04,–15.60). (G) Representative Sholl plots of vehicle (left) and NTG (right) treated mice. (H) Sholl analysis broken up by 0.377 µm/20 pixel distances from the center of the cell showing differences between groups. (I) Sholl analysis revealed a significant decrease in total intersections after chronic NTG treatment. Unpaired t-test. *p=0.0193, t = 2.396, df = 70, 95% CI (−9.825,–0.8977) n = 6 mice/group, six neurons per mouse.

The online version of this article includes the following source data and figure supplement(s) for figure 1:

**Source data 1.** Data *Figure 1B*.
**Source data 2.** Data *Figure 1E*.
**Source data 3.** Data *Figure 1F*.
**Source data 4.** Data *Figure 1H*.
**Source data 5.** Data *Figure 1I*.

*Figure 1 continued on next page*

*Figure 1 continued*

**Figure supplement 1.** Chronic NTG treatment causes cytoarchitectural changes in the Somatosensory Cortex (SCx) and Periaqueductal Gray (PAG) but not the Nucleus Accumbens (Nac) Lumbar Spinal Cord (LSC), or Cervical Spinal Cord (CSC).

**Figure supplement 1—source data 1.** Data *Figure 1—figure supplement 1B*.

**Figure supplement 1—source data 2.** Data *Figure 1—figure supplement 1C*.

**Figure supplement 1—source data 3.** Data *Figure 1—figure supplement 1D*.

**Figure supplement 1—source data 4.** Data *Figure 1—figure supplement 1F*.

**Figure supplement 1—source data 5.** Data *Figure 1—figure supplement 1G*.

**Figure supplement 1—source data 6.** Data *Figure 1—figure supplement 1H*.

**Figure supplement 1—source data 7.** Data *Figure 1—figure supplement 1J*.

**Figure supplement 1—source data 8.** Data *Figure 1—figure supplement 1K*.

**Figure supplement 1—source data 9.** Data *Figure 1—figure supplement 1L*.

**Figure supplement 1—source data 10.** Data *Figure 1—figure supplement 1N*.

**Figure supplement 1—source data 11.** Data *Figure 1—figure supplement 1O*.

**Figure supplement 1—source data 12.** Data *Figure 1—figure supplement 1P*.

**Figure supplement 1—source data 13.** Data *Figure 1—figure supplement 1R*.

**Figure supplement 1—source data 14.** Data *Figure 1—figure supplement 1S*.

**Figure supplement 1—source data 15.** Data *Figure 1—figure supplement 1T*.

processed and analyzed by western blot. ACY-738 treatment resulted in a significant increase in the ratio of acetylated α-tubulin to total tubulin within the trigeminal ganglia (TG), TNC, SCx, NAc, and spinal cord (*Figure 2—figure supplement 1*). A separate group of mice were analyzed by neuronal tracing in the TNC, 4 hr post-ACY-738 (*Figure 2A*). Again, chronic NTG treatment caused a decrease in branch points (*Figure 2B*), combined neurite length (*Figure 2C*), and number of intersections (*Figure 2D–F*). In contrast, treatment with ACY-738 led to a significant increase in these measures in both chronic vehicle and NTG groups (*Figure 2A–F*).

## HDAC6 inhibition reverses NTG-induced allodynia and restores neuronal complexity in a time dependent manner

We next determined if this restored neuronal complexity would affect behavioral outcomes. Mice were treated with chronic intermittent NTG or vehicle for 9 days. On day 10, baseline cephalic allodynia was observed in mice treated chronically with NTG but not vehicle (*Figure 3A*, baselines). Mice were then injected with ACY-738 or vehicle and tested 4, 24, or 48 hr later. ACY-738 significantly reversed cephalic allodynia in NTG treated mice for up to 24 hr post-injection. Mechanical responses in ACY-738 treated animals, not treated with NTG were unaffected (*Figure 3A*, VEH-ACY), suggesting that augmentation of neuronal complexity by HDAC6 inhibitor in a pain-free animal does not alter endogenous pain processing. Interestingly, the half-life of ACY-738 is only 12 min (*Jochems et al., 2014*), thus short-term inhibition of HDAC6 still produced long-lasting behavioral and cytoarchitectural changes.

To explore further the correlation between neuronal complexity and allodynia, we examined the cytoarchitecture of neurons following treatment with ACY-738 at the 24 and 48 hr time points (*Figure 3B*). A separate cohort of animals were treated with NTG/Veh for 9 days and on day 10 received vehicle/ACY-738 and were sacrificed 24 or 48 hr post-injection for neuronal analysis. At 24 hr post-ACY-738, NTG-Veh animals continued to show significantly fewer branch points as compared to the Veh-Veh controls (*Figure 3C*). In contrast, the NTG-ACY group had a significantly higher number of branches compared to NTG-Veh. Interestingly, at 24 hr we no longer observed increased neuronal complexity in the vehicle-ACY-738 group, as was observed at the 4 hr time point (*Figure 2*). Similar results were observed for total neuronal length (*Figure 3D*) and interactions (*Figure 3E*). These data further strengthen the correlation between cytoarchitectural complexity and decreased allodynia, as at both 4 and 24 hr post-injection ACY-738 has anti-allodynic effects.

At 48 hr post-ACY-738 we observed that compared to the VEH-VEH controls, the number of branch points were now significantly lower in both the NTG-vehicle and NTG-ACY groups (*Figure 3F*); and similar results were observed for total neuronal length (*Figure 3G*). Only the NTG-Veh group showed significantly fewer interactions, although there was a trend in the NTG-ACY

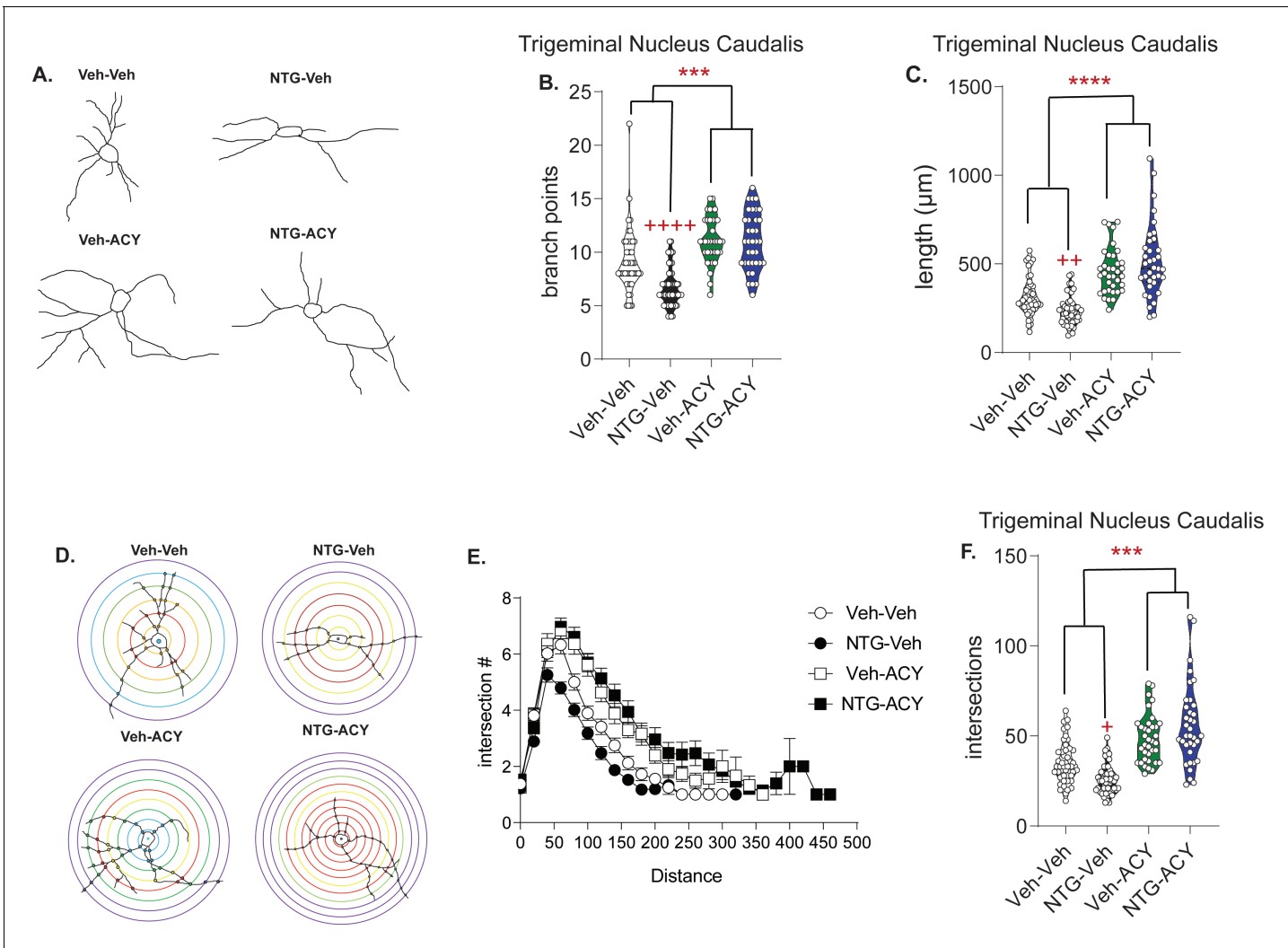

**Figure 2.** Treatment with HDAC6 inhibitor restores blunted neuronal complexity. (**A**) Representative neuron tracing of mice that were chronically treated with vehicle or NTG (10 mg/kg, IP) every other day for 9 days and on day 10 were injected with ACY-738 (50 mg/kg, IP) or vehicle (5% DMSO in 0.9% NaCl, IP) and sacrificed 4 hr later for Golgi staining. (**B**) The number of branch points/neuron were significantly decreased following NTG treatment (NTG-Vehicle); while ACY-738 treated mice showed increased branching regardless of pretreatment. p=0.0061, F = 7.891, 95% CI (0.4864, 2.841) effect of chronic treatment, p=0.0002, F = 14.61, 95% CI (−3.441,−1.086) drug treatment, p=0.0044, F = 8.537, 95% CI (−5.815,−1.106) and interaction, two-way ANOVA and Holm-Sidak post hoc analysis. ***p=0.0002, t = 4.851 effect of ACY-738; ++++p<0.0001, t = 4.947 effect of NTG. (**C**) NTG also decreased total length/neuron, while ACY-738 treatment increased it. p<0.0001, F = 109.8, 95% CI (−667.8,−456.1) effect drug treatment, p=0.0004, F = 12.82, 95% CI (−595.8,−172.3) and interaction, two-way ANOVA and Holm-Sidak post hoc analysis. ****p<0.0001, t = 10.06 effect of ACY-738; ++p=0.0019, t = 3.358 effect of NTG. (**D**) Representative Sholl analysis image of neuronal complexity in the four groups. (**E**) Sholl analysis broken up by 0.377 μm/20 pixel distances reveal differences between NTG and ACY-738 treatment. (**F**) Chronic NTG results in significantly fewer total interactions relative to vehicle treatment. ACY-738 increases total interactions in both vehicle and NTG groups. p<0.0001, F = 59.38, 95% CI (−27.60,−15.96) effect drug treatment, and p=0.0073, F = 8.537, 95% CI (−28.16,−4.874) interaction, two-way ANOVA and Holm-Sidak post hoc analysis. ****p<0.001, t = 3.342 effect of ACY-738; +p=0.0403, t = 2.477 effect of NTG. For all analysis n = 6 mice/group, 6–8 neurons per mouse.

The online version of this article includes the following source data and figure supplement(s) for figure 2:

**Source data 1.** Data *Figure 2B*.
**Source data 2.** Data *Figure 2C*.
**Source data 3.** Data *Figure 2E*.
**Source data 4.** Data *Figure 2F*.
**Figure supplement 1.** ACY-738 increased levels of acetylated α-tubulin.
**Figure supplement 1—source data 1.** Data *Figure 2—figure supplement 1A*.
**Figure supplement 1—source data 2.** Data *Figure 2—figure supplement 1B*.
**Figure supplement 1—source data 3.** Data *Figure 2—figure supplement 1C*.
*Figure 2 continued on next page*

*Figure 2 continued*

**Figure supplement 1—source data 4.** Data *Figure 2—figure supplement 1D*.
**Figure supplement 1—source data 5.** Data *Figure 2—figure supplement 1E*.

group (*Figure 3H*). Taken together, these data show that the anti-allodynic effects of ACY-378 correspond with the times at which it also restores neuronal complexity.

We confirmed that this behavioral effect resulted from changes in HDAC6 inhibition. We first tested two pan-HDAC inhibitors: the well-characterized inhibitor trichostatin A (TSA, *Figure 4A*); and a novel brain-penetrant pan-HDAC inhibitor, RN-73 (*Abdelkarim et al., 2017*; *Figure 4B*). Both significantly reversed chronic NTG-induced allodynia, albeit for a much shorter duration than ACY-738. In contrast, when we tested the Class I, HDAC1 and 2 selective inhibitor, ASV-85 (*Supplementary file 1*), we did not observe any change in NTG-induced chronic allodynia relative to vehicle controls (*Figure 4C*). These data further support our finding that chronic migraine-associated pain can be blocked specifically by HDCA6 inhibition, and that this effect is not likely due to acetylation of histones in the cell nucleus.

Considering the acute effectiveness of ACY-738, we next determined if sustained HDAC6 inhibition could block the establishment of NTG-induced hypersensitivity. In this case, ACY-738/VEH was injected 2 hr before NTG/VEH administration every other day for 9 days. Basal cephalic mechanical thresholds were assessed before drug treatment on days 1, 5, and 9. As seen previously, chronic NTG resulted in the development of a chronic allodynia (*Figure 4D*, VEH-NTG vs VEH-VEH); and concurrent treatment with ACY-738 prevented the development of this allodynia (VEH-NTG vs ACY-NTG). These data demonstrate that chronic HDAC6 inhibition can prevent the development of chronic migraine-associated pain, further supporting the potential of HDAC6 inhibitors as a therapeutic target for migraine.

## HDAC6 mRNA and protein is found ubiquitously in key migraine processing regions

HDAC6 expression is enriched in certain brain regions, such as the dorsal raphe (*Espallergues et al., 2012*); and to the best of our knowledge HDAC6 expression in head pain processing regions is not well characterized. In situ hybridization using RNAScope and immunohistochemical analysis (*Figure 4—figure supplement 1A,B*) revealed abundant expression of HDAC6 transcripts in TG, TNC, and SCx. Gene expression analysis revealed that, of these regions, chronic NTG treatment increased HDAC6 expression in the TG (*Figure 4-figure supplement 1C*), which are the first-order cells regulating cephalic pain processing. Thus, HDAC6 is expressed and regulated dynamically in regions that are critical for migraine-associated pain processing.

## HDAC6 inhibitor results in reduced CSD events

CSD is an electrophysiological property thought to underlie migraine aura. It is mechanistically and etiologically distinct from the NTG model of migraine pain, and reduction of CSD events is a feature of many migraine preventives (*Ayata et al., 2006*). Thus, we examined whether CSD propagation was also affected by HDAC6 inhibition. Briefly, the skull was thinned in an anesthetized animal to reveal the dural vasculature and cortex underneath (*Figure 5A*). Two burr holes were made, and the more rostral was used to continuously drip KCl onto the dura to induce CSD, while local field potentials (LFPs) were recorded from the caudal burr hole. The somatosensory/barrel cortex was targeted, as it is more sensitive to CSD induction (*Bogdanov et al., 2016*). Throughout the 1 hr recording, CSDs were identified by visual shifts in light and sharp decreases in the LFP (*Figure 5B–C*). Pretreatment with ACY-738 resulted in significantly fewer CSD events relative to vehicle controls (*Figure 5D*), indicating that HDAC6 inhibition is also effective in this mechanistically separate migraine model.

## CSD results in decreased neuronal complexity in the somatosensory cortex that is restored by HDAC6 inhibition

We next examined the neuronal complexity of pyramidal neurons within the somatosensory cortex following CSD induction. Sham mice that underwent anesthesia and surgery, but did not receive

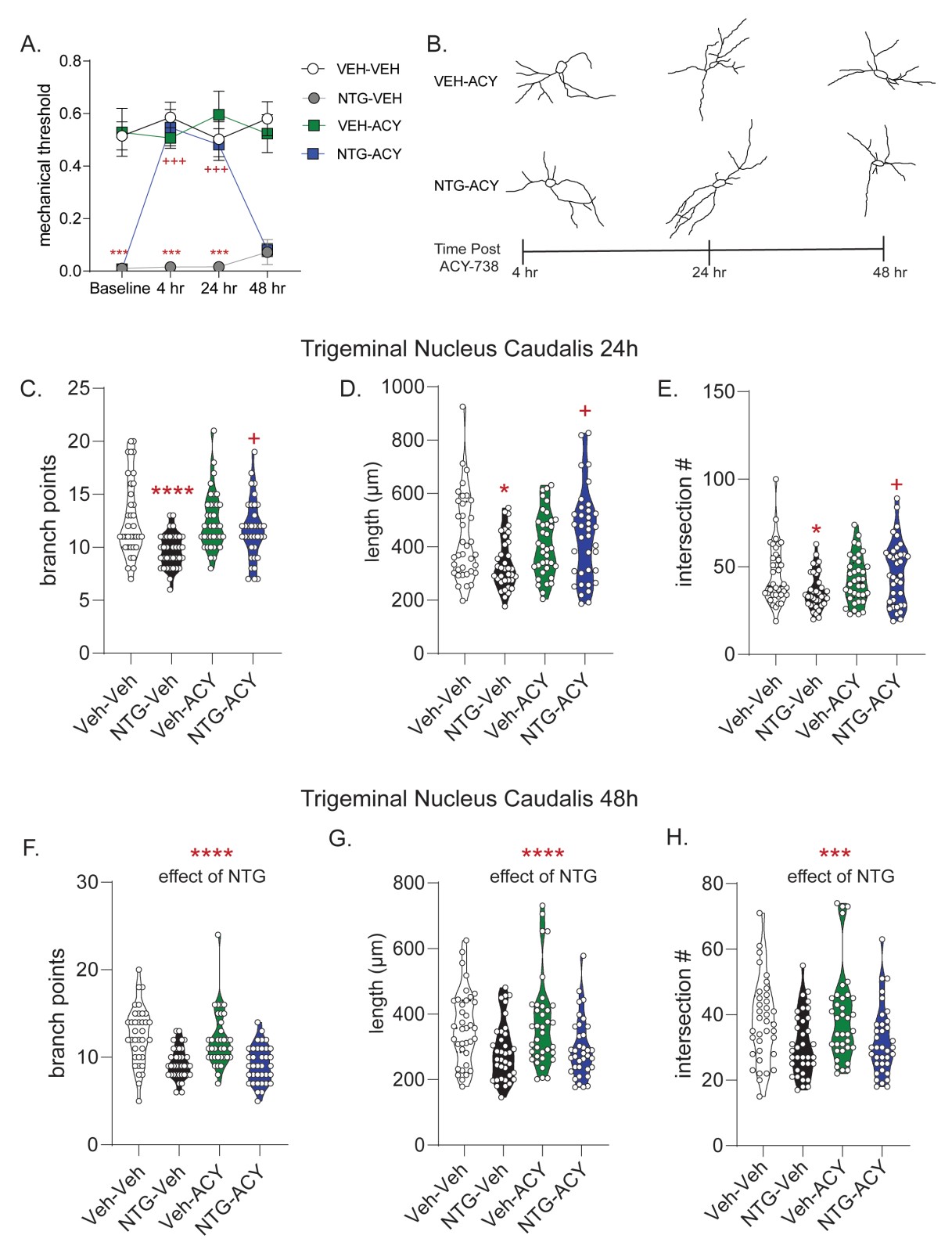

**Figure 3.** ACY-738 reverses established allodynia in a time-dependent manner with neuronal alterations. (**A**) C57Bl/6J mice underwent chronic intermittent NTG/Veh treatment for 9 days, on day 10 basal mechanical thresholds were assessed, and mice were subsequently injected with ACY-738 (50 mg/kg IP) or Vehicle and tested 4, 24, and 48 hr later. Chronic NTG treatment caused severe cephalic allodynia (Baselines); which was significantly inhibited by ACY-738 at 4 hr and 24 hr post-injection. p<0.0001, F = 209.8 chronic treatment, p=0.0002, F = 16.85 drug, p=0.0046, F = 5.174 time, and

*Figure 3 continued on next page*

*Figure 3 continued*

interaction p<0.0001 F = 11.19, three-way RM ANOVA, Holm- Sidak post hoc analysis, ***p<0.001, t = 7.006 NTG-Veh as compared to Veh-Veh at baseline, ***p<0.001 t = 9.126 NTG-Veh as compared to Veh-Veh at 4 hr time point, ***p<0.001 t = 6.736 NTG-Veh as compared to Veh-Veh at 24 hr timepoint; +++p<0.001, t = 7.993 NTG-ACY compared to the NTG-Veh treated mice at 4 hr timepoint, +++p<0.001, t = 6.069; n = 12 mice/group. (B) Representative trigeminal nucleus caudalis neurons from mice treated with VEH-ACY (top) or NTG-ACY (bottom) depicting alterations in cytoarchitecture over the 48 hr time course. In the trigeminal nucleus caudalis, the (C) number of branch points/neuron were significantly decreased following NTG treatment (NTG-Vehicle) an effect not observed following treatment with ACY (NTG-ACY); p<0.0001, F = 17.46, 95% CI (1.024, 2.864) effect of chronic treatment, p=0.0664, F = 3.424, 95% CI (−1.781, 0.05890) drug treatment, p=0.0183, F = 5.701, 95% CI (−4.062,−0.3822) and interaction, two-way ANOVA and Holm-Sidak post hoc analysis. ****p<0.0001, t = 4.643 NTG-Veh compared to Veh-Veh; +p=0.0128, t = 2.997 NTG-Veh compared to NTG-ACY. (D) NTG also decreased total length/neuron, while ACY-738 treatment restored length in chronic NTG-treated animals. p=0.0955, F = 2.817, 95% CI (−226.4, 18.49) effect drug treatment, and p=0.0062, F = 7.717, 95% CI (−588.9,−99.19) interaction, two-way ANOVA and Holm-Sidak post hoc analysis. *p=0.0217, t = 2.897 NTG-Veh compared to Veh-Veh; +p=0.0119, t = 3.151 NTG-Veh compared to NTG-ACY. (E) Chronic NTG resulted in significantly fewer total interactions relative to vehicle treatment, an effect not observed in ACY-738-treated groups. p=0.1278, F = 2.347, 95% CI (−8.780, 1.113) effect drug treatment, p=0.2641, F = 1.257, 95% CI (−2.141, 7.752) chronic treatment, and p=0.0084, F = 7.158, 95% CI (−23.28,−3.495) interaction, two-way ANOVA and Holm-Sidak post hoc analysis. *p=0.040, t = 2.685 Veh-Veh compared to NTG-Veh; +p=0.0205, t = 2.975 NTG-Veh compared to NTG-ACY. For all analyses n = 6 mice/group, six neurons per mouse. At 48 hr, the restorative effects of ACY-738 were no longer present. Chronic NTG treated groups now showed decreased (F) branching. ****p<0.001, F = 46.33, 95% CI (2.138, 3.889) effect of chronic NTG two-way ANOVA. (G) Combined neurite length was also lower in NTG treated groups. ****p<0.0001, F = 17.41, 95% CI (39.94, 111.9) effect of chronic NTG. (H) Chronic NTG groups also showed decreased interactions following Sholl analysis at the 48 hr timepoint. ***p=0.0002, F = 14.99, 95% CI (3.725, 11.50) effect of chronic NTG two-way ANOVA.

The online version of this article includes the following source data for figure 3:

**Source data 1.** Data *Figure 3A*.
**Source data 2.** Data *Figure 3C*.
**Source data 3.** Data *Figure 3D*.
**Source data 4.** Data *Figure 3E*.
**Source data 5.** Data *Figure 3F*.
**Source data 6.** Data *Figure 3G*.
**Source data 7.** Data *Figure 3H*.

KCl, were used as controls. Mice were pretreated with ACY-738 or vehicle, underwent CSD or sham procedure, and were immediately sacrificed for Golgi staining of the SCx (*Figure 6A–B*). In the somatosensory cortex, CSD evoked a significant decrease in branch points (*Figure 6C*) and total length of neurons (*Figure 6D*). In addition, CSD also resulted in a significant reduction in the number of branches in neurons of the TNC (*Figure 6—figure supplement 1*), a region that is known to be activated following CSD events (*Zhang et al., 2011*). In contrast, ACY-738 increased neuronal complexity in the cortex in both sham and CSD groups. Sholl analysis demonstrated a dramatic decrease in neuronal complexity after CSD, while ACY-738 treatment had the opposite effect (*Figure 6E–G*). These results demonstrate that decreased neuronal complexity is also observed in a second, mechanistically distinct model of migraine, and that HDAC6 inhibition can prevent these changes in neuronal cytoarchitecture and decrease CSD events.

## CGRP receptor blockade reverses NTG-induced chronic allodynia and cytoarchitectural alterations

We next sought to determine if migraine-selective therapies could influence neuronal cytoarchitecture; and we tested the small molecule CGRP receptor antagonist, olcegepant, in the chronic NTG model (*Olesen et al., 2004*). Mice developed a sustained allodynia to repeated NTG treatment (*Figure 7A*). On day 10, 24 hr after the final NTG injection, baseline mechanical responses were determined and mice were treated with olcegepant or vehicle. Olcegepant significantly inhibited NTG-induced cephalic allodynia (*Figure 7B*), similar to previously published reports (*Christensen et al., 2019*). Subsequent Golgi analysis of TNC revealed cytoarchitectural alterations in this cohort of animals (*Figure 7C*). As was observed previously, chronic NTG treatment decreased the number of branch points (*Figure 7D*), combined neurite length (*Figure 7E*), and number of intersections using Sholl analysis (*Figure 7F–H*). Interestingly, olcegepant treatment restored neuronal complexity induced by chronic NTG, but had no effect in chronic vehicle-treated mice (*Figure 7C–*

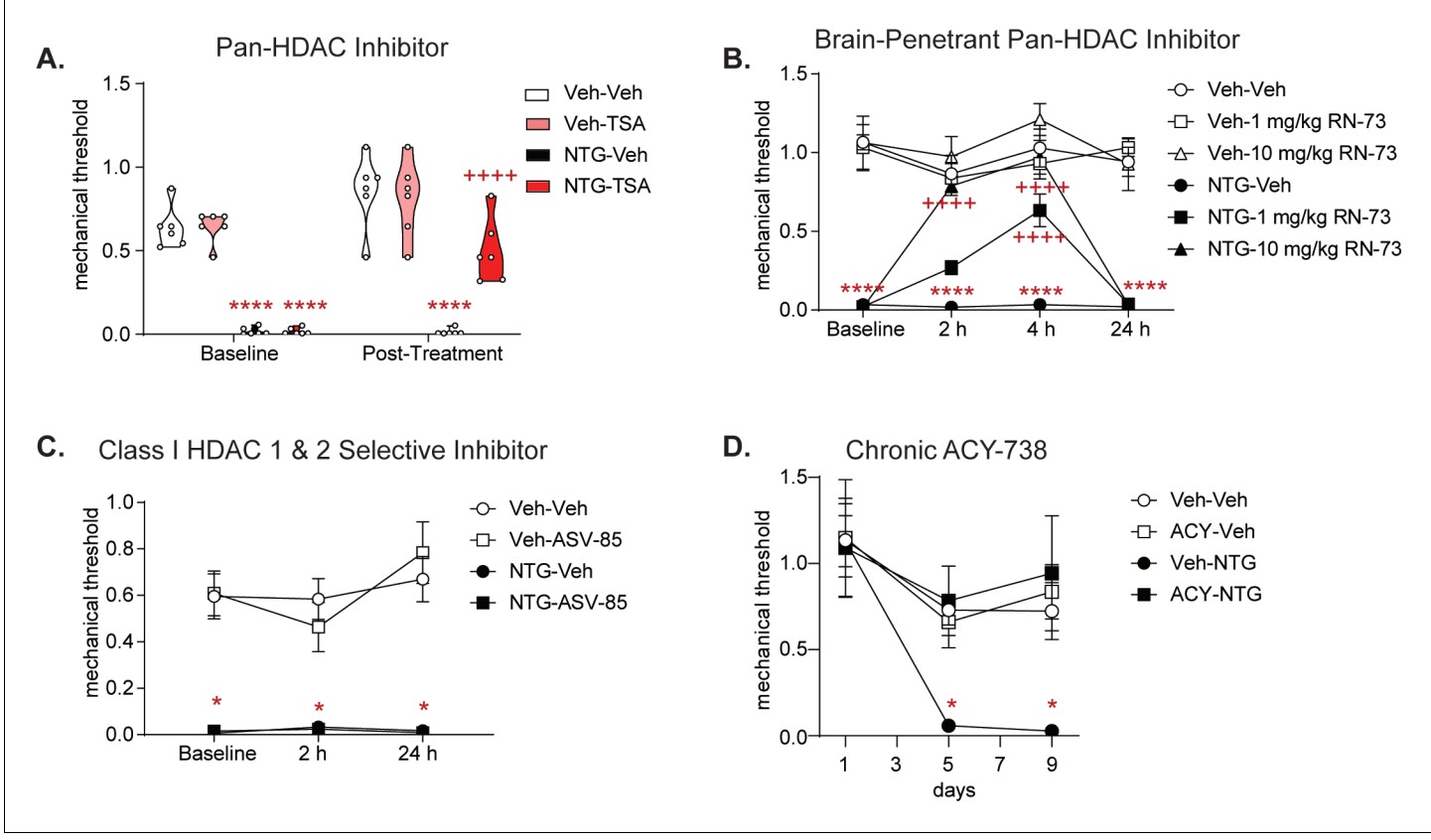

**Figure 4.** Pan-HDAC inhibitors, but not Class I-selective HDAC inhibitor block chronic migraine-associated pain; and repeated ACY-738 can prevent development of NTG induced allodynia. Male and female C57BL/6J mice were treated with chronic intermittent NTG (10 mg/kg IP) or Vehicle for 9 days. On day 10, mice were subsequently tested for baseline responses (Baseline) and then injected with various HDAC inhibitors. Baselines were always lower for NTG-treated mice demonstrating chronic allodynia. Separate groups of mice were tested for each drug. (**A**) Mice were treated with the pan-HDAC6 inhibitor Trichostatin A (TSA, 2 mg/kg IP) or Vehicle (20% DMSO in 0.01 M PBS IP) and subsequently tested 2 hr post-drug. TSA significantly inhibited chronic cephalic allodynia. p<0.0001, F = 271.6 effect of chronic treatment, p=0.0066, F = 9.193 effect of drug, p=0.0001, F = 22.30 time, and p=0.0080, F = 8.688 interaction, Three-way ANOVA and Holm-Sidak post hoc analysis. ****p<0.0001, t = 7.507 NTG-Veh relative to Veh-Veh at baseline ****p<0.0001, t = 7.506 NTG-TSA relative to Veh-Veh at baseline, ****p<0.0001 t = 4.351 NTG-Veh compared to Veh-Veh 2 hr time point. ++++p<0.0001, t = 5.872 relative to NTG-Vehicle at 2 hr time point n = 6–10/group (**B**) Mice were treated with the novel brain-penetrant pan-HDAC inhibitor, RN-73 at 1 or 10 mg/kg (IP) or vehicle (10% DMSO, 10%Tween-80, and 0.9% NaCl). Mice were subsequently tested in the hind-paw region 2, 4, and 24 hr post-treatment. RN-73 at the 10 mg/kg dose had a significant effect at the 2 and 4 hr time point, and the 1 mg/kg dose had a significant effect only at the 4 hr time point compared to the NTG-Veh group. Neither dose of RN-73 produced any effect 24 hr after treatment compared to NTG-Veh. Three-way RM ANOVA and Holm-Sidak post hoc analysis, High Dose-p <0.0001, F = 135.3 effect of treatment, p=0.0009, F = 13.87 drug, p<0.0001, F = 15.01 time and p=0.0002, F = 7.396 interaction; Low Dose- p<0.0001, F = 240.9 effect of treatment, p=0.0780, F = 3.347 drug, p=0.0025, F = 5.183 time and p=0.0002, F = 7.214 interaction, ****p<0.0001, t = 7.858 NTG-Veh relative to Veh-Veh at baseline, ****p<0.0001, t = 6.458 NTG-Veh relative to Veh-Veh at 2 hr time point, ****p<0.0001, t = 7.528 NTG-Veh relative to Veh-Veh at 4 hr time point, ****p<0.0001, t = 7.023 NTG-Veh relative to Veh-Veh at 24 hr time point; ++++p<0.0001, t = 5.867 NTG- RN-73 high dose relative to NTG-Vehicle at 2 hr point, ++++p<0.0001, t = 7.138 NTG- RN-73 high dose relative to NTG-Vehicle at 4 hr time point, ++++p<0.0001, t = 5.480 NTG- RN-73 low dose relative to NTG-Vehicle at 4 hr time point n = 8/group. (**C**) Following chronic NTG/VEH treatment, mice were treated with the Class I specific HDAC inhibitor, ASV-85 (1 mg/kg IP) or Vehicle (6.25% DMSO, 5.625% Tween- 80, and 0.9% NaCl, IP) and had subsequent mechanical thresholds taken at 2 and 24 hr post-treatment. ASV-85 failed to inhibit NTG-induced pain. NTG-ASV-85 and NTG-Veh treated mice both were significantly different than the Vehicle control groups at both 2 and 24 hr time points, Three-way ANOVA and Holm-Sidak post hoc analysis p<0.0001, F = 118.1 effect of chronic treatment, *p=0.0264, q = 8.451 NTG-ASV compared to Veh-Veh at Baseline, *p=0.0196, q = 8.860 Veh-Veh compared to NTG-ASV at 2 hr time point, *p=0.0161 q = 9.568 Veh-Veh compared to NTG-ASV at 24 hr time point treated mice at same time point. n = 6/group (**D**) Mice were concurrently treated with ACY-738 which prevented development of basal allodynia. p=0.0309, F = 5.672 effect of NTG/Veh treatment, p=0.0663, F = 3.922 effect of ACY/Veh, p=0.0073, F = 6.947 effect of Time, p=0.0253, F = 6.617 effect of interaction of NTG/Veh and ACY/Veh, *p=0.0329, t = 3.727 NTG-Veh day 5 compared to Veh-Veh day 1, *p=0.0250, t = 3.834 NTG-Veh day 5 compared to Veh-Veh day 9 n = 6/group.

The online version of this article includes the following source data and figure supplement(s) for figure 4:

**Source data 1.** Data *Figure 4A*.
**Source data 2.** Data *Figure 4B*.
*Figure 4 continued on next page*

*Figure 4 continued*

**Source data 3.** Data *Figure 4C*.
**Source data 4.** Data *Figure 4D*.
**Figure supplement 1.** HDAC6 is expressed in migraine-processing regions and is dynamically regulated.
**Figure supplement 1—source data 1.** Data *Figure 4—figure supplement 1C*.

*H*). These data reinforce the concept that altered neuronal complexity could be a feature of chronic

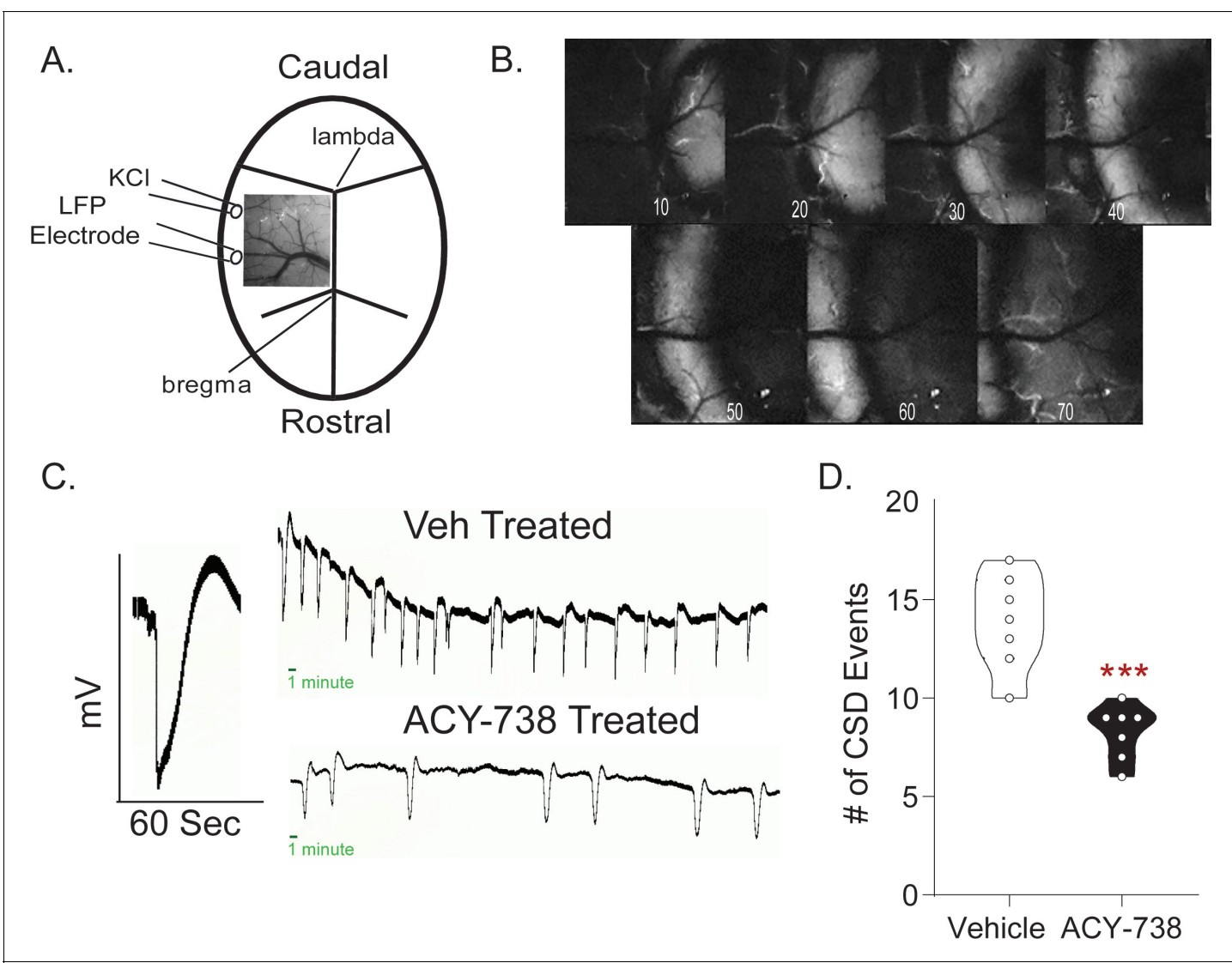

**Figure 5.** ACY-738 reduces cortical spreading depression events. (**A**) Schematic of the thinned skull preparation used to visualize CSD and placement of KCl infusion and LFP recording. (**B**) Image sequence shows the wave of change in reflectance associated with a CSD event. (**C**) Representative tracing of a single CSD event of voltage change versus time. Representative line tracing of CSDs in a Vehicle (Top) vs. ACY-738 (Bottom) treated mouse over a 1 hr period. (**D**) Animals pretreated with ACY-738 (50 mg/kg IP) 3 hr before CSD recordings began showed a significant reduction in the average number of CSD events recorded over an hour. Unpaired t-test. ***p<0.001, t = 5.307, df = 12, 95% CI (−7.859,–3.284) n = 7/group.
The online version of this article includes the following source data for figure 5:

**Source data 1.** Data *Figure 5D*.

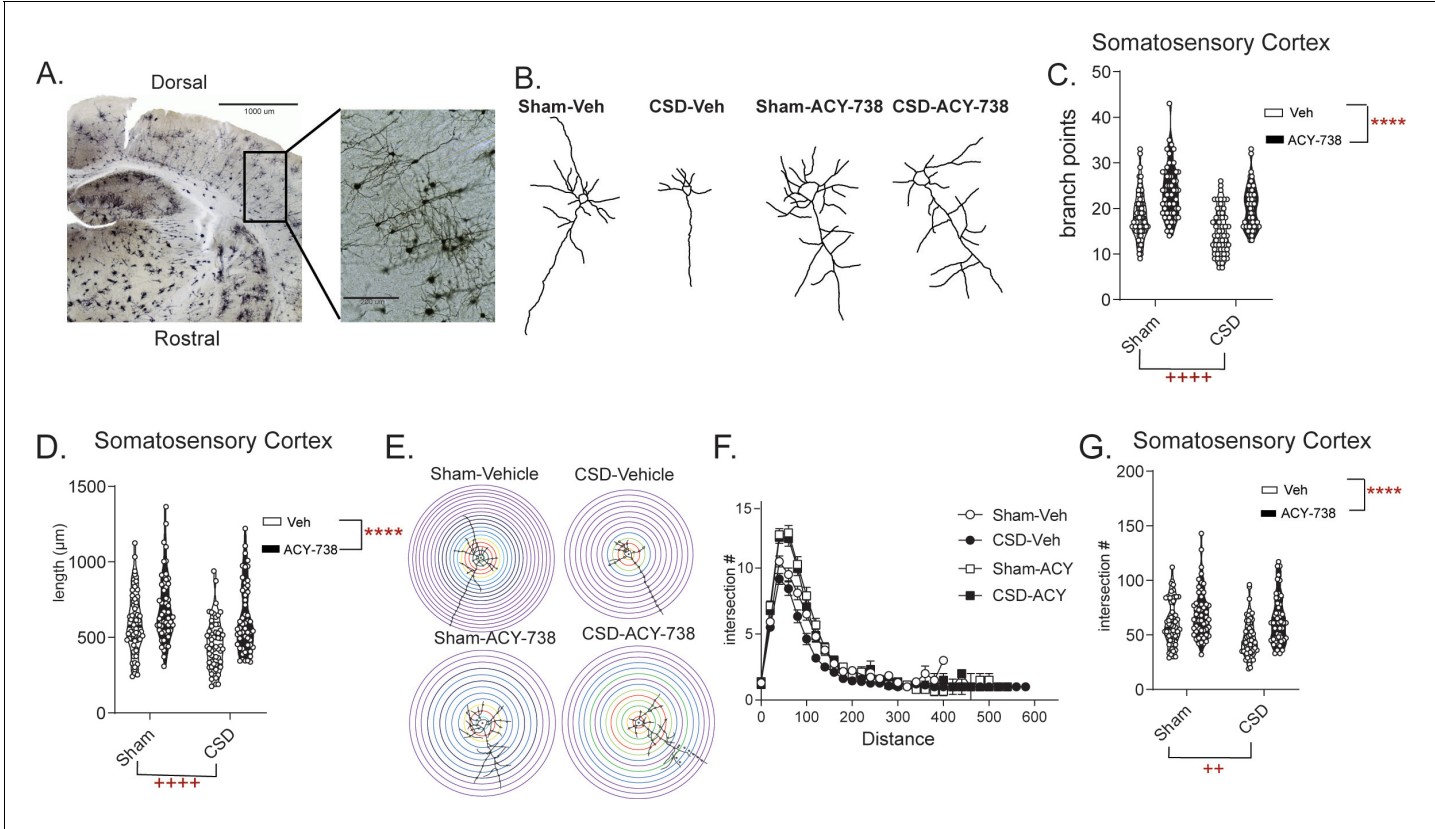

**Figure 6.** CSD induces decreased neuronal complexity that is prevented by treatment with ACY-738. (**A**) Representative image of Golgi stained sensory/barrel cortex at 4x (left) and 20x (right). (**B**) Representative neuronal tracing for mice that underwent pretreatment with Vehicle or ACY-738 and underwent Sham or CSD procedures. (**C**) Analysis of number of branch points/neuron reveal a significant effect of CSD and of ACY-738. Two-way ANOVA analysis. ++++p<0.0001, F = 27.80, 95% CI (2.274, 4.989) effect of CSD; ****p<0.0001, F = 60.66, 95% CI (−6.722,–4.007) effect of ACY-738. (**D**) Neurons were further analyzed for length per neuron and CSD significantly decreased overall length, while ACY-738 significantly increased length. Two-way ANOVA. ++++ p<0.0001, F = 16.48, 95% CI (56.14, 162.1) effect of CSD, ****p<0.0001, F = 25.18, 95% CI (−187.9,–81.91) effect of ACY-738 (**E**) Representative Sholl analysis plot of a neuron demonstrating CSD reduces and ACY-738 increases neuronal complexity. (**F**) Sholl analysis broken up by 0.377 μm/20 voxel distances showing differences between groups. (**G**) Sholl analysis revealed a significant decrease in total intersections after CSD compared to Sham mice; and pretreatment with ACY-738 increased total intersections compared to vehicle-treated groups. Two-way ANOVA ++p=0.0011, F = 10.87. 95% CI (3.601, 14.30) effect of CSD, ****p<0.0001, F = 23.98, 95% CI (−18.64,–7.944) effect of ACY-738. n = 6 mice/group, nine neurons/mouse.

The online version of this article includes the following source data and figure supplement(s) for figure 6:

**Source data 1.** Data *Figure 6C*.

**Source data 2.** Data *Figure 6D*.

**Source data 3.** Data *Figure 6F*.

**Source data 4.** Data *Figure 6G*.

**Figure supplement 1.** Cortical spreading depression (CSD) results in blunted neuronal complexity in the trigeminal nucleus caudalis (TNC).

**Figure supplement 1—source data 1.** Data *Figure 6—figure supplement 1A*.

**Figure supplement 1—source data 2.** Data *Figure 6—figure supplement 1B*.

**Figure supplement 1—source data 3.** Data *Figure 6—figure supplement 1C*.

migraine, and that restoration of these changes may be a marker of effective migraine treatment.

## Discussion

Our results indicate that in models of chronic migraine and aura there is a dysregulation of cellular plasticity resulting in decreased neuronal complexity. We found that following the establishment of chronic cephalic allodynia in the NTG model of migraine-associated pain, there was a decrease in the number of branch points, combined neurite length, and interactions of neurons within the TNC,

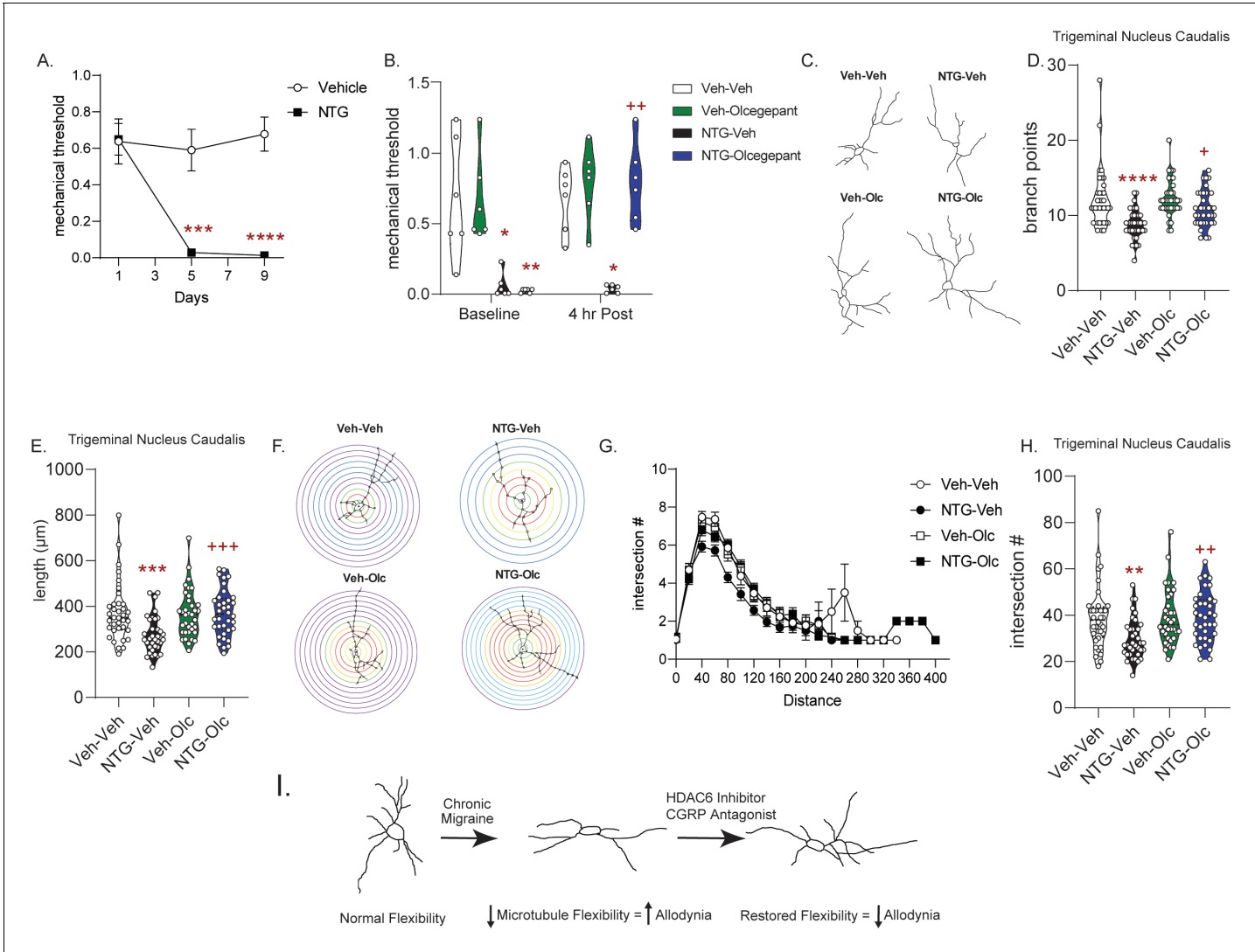

**Figure 7.** Treatment with the CGRP receptor antagonist, olcegepant, blocks NTG-induced chronic allodynia and reverses blunted cytoarchitecture. (**A**) Periorbital mechanical thresholds were accessed prior to Vehicle/NTG administration on days 1, 5, and 9. NTG produced cephalic allodynia; p=0.0002, F = 19.16, 95% CI (0.2129, 0.5964) effect of chronic treatment, p<0.0001, F = 14.25 time, and p<0.0001, F = 14.05 interaction, two-way RM ANOVA and Holm-Sidak post hoc analysis. ***p=0.0009, t = 4.904 Veh compreared to NTG Day 5, ****p<0.0001, t = 7.110 Veh compared to NTG Day 9 n = 12/ group. (**B**) Mice were treated with olcegepant (1 mg/kg, IP) or vehicle (0.9% NaCl) and tested 4 hr later. Olcegepant significantly reversed chronic cephalic allodynia. p=0.0002, F = 32.12 effect of chronic treatment, p=0.0014, F = 19.29 drug, p=0.0172, F = 8.126 time, and p=0.0007, F = 23.40 interaction of time and Veh/Olc. Three-way ANOVA and Holm-Sidak post hoc analysis. *p=0.0137, t = 3.857 NTG-Veh compared to Veh-Veh at Baseline, **p=0.0024, t = 4.798 NTG-Olc compared to Veh-Veh at baseline, *p=0.0110, t = 3.980 NTG-Veh compared to Veh-Veh at 4 hr time point, + +p=0.0026 NTG-Veh relative to NTG-Olc at the 4 hr time point n = 6/group. (**C**) Representative neuron tracing of mice that were chronically treated with vehicle/NTG every other day for 9 days and on day 10 were treated with olcegepant or vehicle and sacrificed 4 hr later for Golgi staining. (**D**) The number of branch points/neuron were significantly decreased following NTG treatment (NTG-Vehicle); while NTG-olcegepant treated mice showed increased branching compared to the NTG-Vehicle treatment alone. p<0.0001, F = 26.82, 95% CI (1.477, 3.301) effect of chronic treatment, p=0.0236, F = 5.237, 95% CI (−1.967,−0.1437) drug treatment, and p=0.0425, F = 4.193, 95% CI (−3.713,−0.06510) interaction, two-way ANOVA and Holm-Sidak post hoc analysis. ****p<0.0001, t = 5.110 NTG-Veh compared to Veh-Veh, +p=0.0104, t = 3.066 NTG-olcegepant compared to NTG-Veh. (**E**) NTG also decreased total length/neuron; while NTG-olcegepant-treated mice showed restored length compared to the NTG-Vehicle treatment alone p=0.0068, F = 7.541, 95% CI (13.40, 82.2) effect of NTG treatment, p=0.0098, F = 6.853, 95% CI (−80.08,−11.17) drug treatment, and p=0.0032, F = 9.009, 95% CI (−173.5,−35.71) interaction, two-way ANOVA and Holm-Sidak post hoc analysis. ***p=0.0005, t = 4.064 NTG-Veh compared to Veh-Veh, +++p=0.0009, t = 3.793 NTG-olcegepant compared to NTG-Veh. (**F**) Representative Sholl analysis image of neuronal complexity in the four groups. (**G**) Sholl analysis broken up by 0.377 μm/20 pixel distances reveal differences between NTG-Veh treatment relative to the other groups. (**H**) NTG-Veh results in significantly fewer total interactions relative to the Veh-Veh treatment and NTG-olcegepant mice had significantly more total interactions compared to NTG-Veh p=0.0210, F = 5.447, 95% CI (0.6772, 8.184) effect chronic treatment, p=0.0127, F = 6.371, 95% CI (−8.545,−1.038) drug treatment, and p=0.0143, F = 6.151, 95% CI (−16.92,−1.910) interaction, two-way ANOVA and Holm-Sidak post hoc analysis. **p=0.0039, t = 3.404 NTG-Veh compared

*Figure 7 continued on next page*

*Figure 7 continued*

to Veh-Veh; ++p=0.0039, t = 3.538 NTG-Veh compared to NTG-olcegepant. For all analysis n = 6 mice/group, six neurons per mouse. (I) Schematic summary of findings. Endogenously there is a balance of acetylated and deacetylated α-tubulin which regulates optimal neuronal complexity. In the case of chronic migraine, there is a disbalance resulting in decreased neuronal complexity. HDAC6 or CGRP receptor inhibition restores tubulin dynamics and neuronal complexity and correspondingly decreases chronic migraine-associated symptoms.

The online version of this article includes the following source data for figure 7:

**Source data 1.** Data *Figure 7A*.
**Source data 2.** Data *Figure 7B*.
**Source data 3.** Data *Figure 7D*.
**Source data 4.** Data *Figure 7E*.
**Source data 5.** Data *Figure 7G*.
**Source data 6.** Data *Figure 7H*.

PAG, and somatosensory cortex. With this newly discovered phenomenon we sought to mitigate this decrease through inhibition of HDAC6 which we found to reciprocally restore neuronal complexity and inhibit allodynia. We found that the cytoarchitectural changes were not just induced by NTG but were also prominent following CSD. Reduction in neuronal complexity was also observed in this model of migraine aura, and again HDAC6 inhibition restored neuronal plasticity and decreased the number of CSD events. The latter effect is a hallmark of migraine preventive drugs. Furthermore, we found that a migraine specific treatment, CGRP receptor inhibition, also restored cytoarchitectural changes. Together our results demonstrate a novel mechanism of chronic migraine and reveal HDAC6 as a novel therapeutic target for this disorder (*Figure 7I*).

We used the NTG model in this study, as it is a well-validated model of migraine (*Demartini et al., 2019*). NTG is a known human migraine trigger and is used as a human experimental model of migraine (*Schytz et al., 2010*). Similar to humans, NTG produces a delayed allodynia in mice (*Bates et al., 2010*), as well as photophobia and altered meningeal blood flow (*Markovics et al., 2012*; *Greco et al., 2011*). Chronic intermittent administration of NTG is used to model chronic migraine (*Pradhan et al., 2014a*; *Farajdokht et al., 2018*; *Long et al., 2018*; *Christensen et al., 2019*; *Zhang et al., 2020*). Compared to humans, much higher doses of NTG are required to produce allodynia. However, NTG-induced hypersensitivity in mice is inhibited by migraine-specific medications, such as sumatriptan (*Bates et al., 2010*; *Pradhan et al., 2014a*; *Pradhan et al., 2014b*) and CGRP targeting drugs (*Christensen et al., 2019*), as well as the migraine preventives propranolol and topiramate (*Tipton et al., 2016*; *Greco et al., 2018*). Further, mice with human migraine gene mutations are more sensitive to NTG (*Brennan et al., 2013*). Systemic administration of NTG also causes cellular activation throughout nociceptive pathways including in the TNC and brainstem (*Tassorelli and Joseph, 1995a*; *Tassorelli and Joseph, 1995b*; *Ramachandran et al., 2012*; *Greco et al., 2018*). Correspondingly, we also observed changes in neuronal complexity in the TNC, as well as in the PAG and somatosensory cortex, regions heavily involved in pain processing. Alterations in these regions could contribute to allodynia or interictal sensitivity observed in chronic migraine patients. Previous studies have shown an increase in TNC activity in headache models (*Oshinsky and Luo, 2006*; *Akerman et al., 2013*). While our data show an overall decrease in complexity within these neurons, they do not necessarily contradict these previous findings. For example, decreased neuronal flexibility could encourage the strengthening of excitatory synapses and/or prevent the formation of inhibitory synapses, as the cell is in a more fixed state. Within the TNC, there are both inhibitory and excitatory neuronal populations; and future studies will determine which populations are altered in migraine models and how these changes directly affect neuronal activity.

Alterations in neuronal complexity in response to NTG appeared to be limited to brain regions involved in pain processing. We did not observe any alterations in the nucleus accumbens which is commonly associated with reward and motivation. RNA-Seq and proteomic experiments from our lab have also revealed that the nucleus accumbens has very different responses to chronic NTG relative to parts of the trigeminovascular system (*Jeong et al., 2018*; *Krishna et al., 2019*). We also observed no change in neuronal cytoarchitecture in the lumbar spinal cord, a region largely involved in peripheral but not cephalic pain processing. Importantly, we found no cytoarchitectural alterations

in the cervical spinal cord, a region that is also involved in head/neck pain processing. These data suggest that decreased neuronal complexity in response to migraine states may be limited to central sites that regulate headache and pain processing.

We also observed decreased neuronal complexity in CSD, a migraine model mechanistically distinct from NTG. CSD is thought to underlie migraine aura and reflects changes in cortical excitability associated with the migraine brain state (*Charles and Baca, 2013*; *Brennan and Pietrobon, 2018*). Previous studies also support the idea of cytoarchitectural alterations accompanying spreading depression/depolarization events and focused mainly on dendritic morphology. Neuronal swelling (*Takano et al., 2007*) and dendritic beading (*Steffensen et al., 2015*) were observed following spreading depression events. CSD also resulted in alterations in dendritic structure (*Takano et al., 2007*; *Eikermann-Haerter et al., 2015*) and volumetric changes (*Takano et al., 2007*). Further, Steffensen et al. showed decreased microtubule presence in dendrites following spreading depression in hippocampal slices, again implying alterations in cytoarchitectural dynamics (*Steffensen et al., 2015*). Microtubules have been shown to disassemble in response to increased intracellular calcium (*Schliwa et al., 1981*); and the increased calcium influx induced by spreading depolarization (*Basarsky et al., 1998*) may facilitate this breakdown. Tubulin acetylation is associated with increased flexibility and stability of microtubules (*Xu et al., 2017*). One way in which HDAC6 inhibitors could attenuate CSD is through increased tubulin acetylation, thus counteracting microtubule disassembly produced by CSD events. In addition, CSD waves pass through the neuron in phases, from apical dendrites, to somatodendritic sites, and finally to proximal dendrites (*Pietrobon and Moskowitz, 2014*). Along with preventing the dendritic alterations that occur in response to calcium influx, it is possible that HDAC6 inhibition could redistribute or disturb the phasic movement of CSD events, which may decrease and/or elongate CSD events. Furthermore, multiple reports indicate that CSD can activate the trigeminovascular complex, and evoke cephalic allodynia in rodents (*Bolay et al., 2002*; *Fioravanti et al., 2011*; *Zhang et al., 2012*; *Noseda and Burstein, 2013*; *Melo-Carrillo et al., 2017*; *Filiz et al., 2019*). We also observed decreased neurite branching in the TNC following CSD, further linking CSD to head pain processing. Combined, these data suggest that CSD has an impact on neuronal morphology that contributes to migraine pathophysiology.

Proper acetylation of microtubules is necessary for a variety of cellular functions including appropriate neurite branching (*Gallo, 2011*), cell response to injury (*Gallo, 2011*), mitochondrial movement (*Braun et al., 2011*; *Jin et al., 2017*), anchoring of kinesin for microtubule mediated transport (*Gibbs et al., 2015*) and regulation of synaptic G protein signaling (*Schappi et al., 2014*; *Singh et al., 2018*; *Singh et al., 2020*). Disruption of this process can have significant physiological effects. Knockout of the α-tubulin acetylating enzyme, α-TAT1, in peripheral sensory neurons results in profound deficits in touch (*Morley et al., 2016*). Further, Charcot-Marie-Tooth (CMT) disease is a hereditary axonopathy that affects peripheral nerves resulting in damage to both sensory and motor function. Mouse models of CMT reveal deficits in mitochondrial transport in the dorsal root ganglia (DRG) due to reduced α-tubulin acetylation, and HDAC6 inhibition ameliorate CMT-associated symptoms (*d'Ydewalle et al., 2011*; *Benoy et al., 2018*). We observed that ACY-738 broadly increased neuronal complexity, including in vehicle and sham controls. However, we did not see any alterations in mechanical thresholds or CSD events in response to this upregulation in these control groups. Furthermore, this effect of ACY-738 appeared to be short lasting and was no longer present at the 24 hr timepoint. Other groups using ACY-738 or other HDAC6 inhibitors also did not observe general disruption in mechanical or temperature sensitivity with HDAC6 inhibition (*Krukowski et al., 2017*; *Van Helleputte et al., 2018*; *Ma et al., 2019*; *Sakloth et al., 2020*). Additionally, constitutive knockout of HDAC6 produces viable offspring with few phenotypic changes (*Zhang et al., 2008*). These findings suggest that HDAC6 inhibition does not appear to generally cause a loss of sensation, but in a migraine state, where mechanical responses are decreased, they can have anti-allodynic effects.

While these results are the first of their kind to demonstrate cytoarchitectural changes in models of chronic migraine, alterations in neuronal plasticity have been described previously in models of neuropathic pain. A mouse model of chronic constriction injury of the sciatic nerve reduced neurite length in GABA neurons within lamina II of the spinal cord (*Zhang et al., 2018*). Another group observed that following spared nerve injury, there were decreases in the number of branches and neurite length of hippocampal neurons but increases in spinal dorsal horn neurons (*Liu et al., 2017*).

These studies, along with our results, suggest that adaptations in response to chronic pain can culminate in alteration of neuronal cytoarchitecture within the central nervous system.

HDAC6 inhibitors have been studied in other models of pain. They were shown to effectively reduce chemotherapy-induced allodynia following treatment with vincristine (*Van Helleputte et al., 2018*) or cisplatin (*Krukowski et al., 2017*) in mice. In addition, both groups found that chemotherapy blunted mitochondrial transport in sensory neurons, an effect that was restored by HDAC6 inhibition. Another study also found that HDAC6 inhibitors were effective in models of inflammatory and neuropathic pain (*Sakloth et al., 2020*). Together, these studies highlight the importance of cytoarchitectural dynamics in relation to pain sensation and the ability of HDAC6 inhibition to promote relief from allodynia/hyperalgesia.

We chose to focus on the role of HDAC6 in tubulin acetylation and microtubule dynamics, as we observed changes in neuronal complexity in migraine models. However, HDAC6 also regulates Hsp90 and cortactin (*Valenzuela-Fernández et al., 2008*). HDAC6 deacetylates Hsp90, which plays an important role in glucocorticoid receptor maturation and adaptation to stress (*Kovacs et al., 2005*). A previous study showed that in social defeat stress HDAC6 knockout or inhibition decreased Hsp90-glucocorticoid receptor interaction and subsequent glucocorticoid signaling, thus encouraging resilience (*Espallergues et al., 2012*). In line with these findings, HDAC6 inhibitors also show antidepressant-like effects (*Covington et al., 2009*; *Jochems et al., 2014*), and membrane-associated acetylated tubulin is decreased in humans with depression (*Singh et al., 2020*). Further, HDAC6 also directly deacetylates cortactin, a protein that regulates actin-dependent cell motility (*Zhang et al., 2007*). Future studies will explore the contribution of these other mechanisms by which HDAC6 may impact neuronal complexity in migraine.

We investigated whether a current migraine treatment strategy, CGRP receptor inhibition, could ameliorate the cytoarchitectural changes induced by chronic migraine-associated pain. We found a good correlation between the anti-allodynic effects of olcegepant and its ability to restore neuronal complexity in the chronic NTG model. In contrast to ACY-738, olcegepant had no effect on vehicle-treated mice and only recovered, but did not increase neuronal branching, length, or intersections in the NTG-treated group. Considering that CGRP receptors are not known to directly affect microtubule dynamics, these results suggest that there are multiple ways through which migraine therapies can affect neuronal plasticity. Previous studies have shown that activation of various G$\alpha$ subunits, including $\alpha_i$, $\alpha_o$, $\alpha_s$, can inhibit microtubule assembly (*Roychowdhury et al., 1999*). Therefore, it is possible that an increase in CGRP, which was found to be present following NTG (*Greco et al., 2018*; *Moye et al., 2021*), could result in altered microtubule assembly. Inhibition of the CGRP receptor could therefore reverse this process allowing for elaboration of microtubules. Olcegepant was previously shown to poorly cross the blood brain barrier; and it may result in cytoarchitectural changes in the central nervous system by blocking nociceptive signals from the periphery, resulting in upstream changes in the TNC and other central regions. Further, the effect of olcegepant along with the finding that NTG did not alter complexity in the spinal cord or nucleus accumbens, help to confirm that the changes in neuronal cytoarchitecture following NTG are associated with migraine mechanisms. This study suggests a possible mechanism in which recovered neuronal complexity is a marker of effective migraine medication.

Our results reveal a novel cytoarchitectural mechanism that may underlie chronic migraine and imply that this disorder results from attenuation of neurite outgrowth and branching. Human imaging studies reveal decreased cortical thickness (*Magon et al., 2019*), and gray matter reductions in the insula, anterior cingulate cortex, and amygdala of migraine patients (*Valfrè et al., 2008*). Interestingly, a significant correlation was observed between gray matter reduction in anterior cingulate cortex and frequency of migraine attacks (*Valfrè et al., 2008*). These structural changes could reflect decreased neuronal complexity in combination with other factors. We propose that strategies targeted toward pathways regulating neuronal cytoarchitecture may be an effective approach for the treatment of chronic migraine. Our results suggest that HDAC6 inhibitors may restore cellular adaptations induced by chronic disease states but may not otherwise affect healthy physiological function; and such compounds could contribute to the migraine therapeutic armamentarium.

# Materials and methods

**Key resources table**

| Reagent type (species) or resource | Designation | Source or reference | Identifiers | Additional information |
|---|---|---|---|---|
| Strain, strain background (Mouse Male and Female) | C57BL/6J | Jackson Laboratories | RRID:IMSR_JAX:000664 | |
| Antibody | Rabbit polyclonal anti-HDAC6 antibody | Tso-Pang Yao Duke University | | (1:500) |
| Antibody | Alexa Fluor 555 Donkey polyclonal anti-Rabbit antibody | Life Technologies | RRID:AB_162543 | (1:2000) |
| Antibody | Mouse monoclonal anti-acetyl-α-tubulin antibody | Sigma | RRID:AB_2819178 (Sigma Clone 6-11B1) | |
| Antibody | HRP-linked goat polyclonal anti-mouse antibody IgG | Jackson Immuno Research | RRID:AB_10015289 | |
| Commercial assay or kit | RNeasy Plus mini kit | Quiagen | | |
| Chemical compound, drug | Nitroglycerin | American Reagent | | Purchased in 30% alcohol, 30% propylene glycol, and water solution |
| Chemical compound, drug | ACY-738 | Acetylon | | 5% DMSO saline solution |
| Chemical compound, drug | Trichostatin A | Sigma | T8552 | |
| Chemical compound, drug | Olcegepant | Tocris | BIBN 4096 | |
| Software, algorithm | Simple Neurite Tracer | | | |

## Animals

Experiments were performed on adult male and female C57BL/6J mice (Jackson Laboratories, Bar Harbor, ME. USA) weighing 20–30 g. Mice were group housed in a 12 h-12h light-dark cycle, where the lights were turned on at 07:00 and turned off at 19:00. Food and water were available ad libitum. All experiments were conducted in a blinded fashion by 1–3 experimenters. Weight was recorded on each test day for all experiments. All experimental procedures were approved by the University of Illinois at Chicago Office of Animal Care and Institutional Biosafety Committee, in accordance with Association for Assessment and Accreditation of Laboratory Animal Care International (AAALAC) guidelines and the Animal Care Policies of the University of Illinois at Chicago. All results are reported according to Animal Research: reporting of In Vivo Experiments (ARRIVE) guidelines. No adverse effects were observed during these studies, and all animals were included in statistical analysis.

## Sensory sensitivity testing

Different groups of animals were used for each experiment. Mice were counter-balanced into groups following the first basal test for mechanical thresholds. Mice were tested in a behavior room, separate from the vivarium, with low light (~35–50 lux) and low-noise conditions, between 09:00 and 16:00. Mice were habituated to the testing racks for 2 days before the initial test day, and on each subsequent test days were habituated for 20 min before the first measurement. The cephalic region was tested throughout this study, except for the RN-73 experiment, where the hind paw was tested. For cephalic measures mice were tested in four oz paper cups. The periorbital region caudal to the eyes and near the midline was tested. For experiments testing peripheral mechanical responses, the intraplantar region of the hindpaw was assessed. Testing of mechanical thresholds to punctate mechanical stimuli was tested using the up-and-down method. The selected region of interest was stimulated using a series of manual von Frey hair filaments (bending force ranging from 0.008 g to 2 g). A response of the head was defined as shaking, repeated pawing, or cowering away from the filament. In the hind paw, a response was lifting of the paw, shaking, or licking the paw after stimulation. The first filament used was 0.4 g. If there was no response a heavier filament (up) was used, and if there was a response a lighter filament (down) was tested. The up-down pattern persisted for

four filaments after the first response. To decrease bias in testing, researchers were blinded to treatment groups at time of testing. While the same researcher performed both the mechanical threshold testing and injections these measures were recorded in different places and at separate time points.

## Nitroglycerin model of chronic migraine

Nitroglycerin (NTG) was purchased at a concentration of 5 mg/ml, in 30% alcohol, 30% propylene glycol and water (American Reagent, NY, USA). NTG was diluted on each test day in 0.9% saline to a concentration of 1 mg/ml for a dose of 10 mg/kg. Mice were administered NTG or vehicle every other day for 9 days. Animals used in cephalic experiments were tested on days 1, 5, and 9. On test days a basal threshold was measured then animals were treated with either NTG or vehicle and then put back in the testing racks and subsequently tested 2 hr later for the post-treatment effect.

## Cortical spreading depression model

The procedure for the cortical spreading depression (CSD) model is based on previously published work (*Chen and Ayata, 2017*) that is commonly used to screen potential migraine preventives and further used in our own work (*Pradhan et al., 2014b*; *Dripps et al., 2020*; *Bertels et al., 2021*). Mice were grouped into sham and CSD groups and then further subdivide into ACY-738 (50 mg/kg, IP) or vehicle (i.e. Sham-ACY, Sham-Veh, CSD-ACY, CSD-Veh). To make the thinned skull cortical window, mice were anesthetized with isoflurane (induction 3–4%; maintenance 0.75% to 1.25%; in 67% $N_2$ / 33% $O_2$) and placed in a stereotaxic frame on a homoeothermic heating pad. Core temperature (37.0 ± 0.5℃), non-peripheral oxygen saturation (~ 99%), heart rate, and respiratory rate (80–120 bpm) were continuously monitored (PhysioSuite; Kent Scientific Instruments, Torrington, CT, USA). Mice were frequently tested for tail and hind paw reactivity to ensure that the anesthesia plane was maintained.

To verify CSD events, optical intrinsic signal (OIS) imaging was primarily used and electrophysiological recordings were recorded as previously described (*Pradhan et al., 2014b*). Briefly, following anesthesia, the skin from the skull was detached and a rectangular region of ~2.5×3.3 mm$^2$ (~0.5 mm from sagittal, and ~1.4 from coronal and lambdoid sutures) of the right parietal bone was thinned to transparency with a dental drill (Fine Science Tools, Inc, Foster City, CA, USA). Mineral oil application improved transparency of cortical surface parenchyma and vasculature for video recording. A green LED (530 nm) illuminated the skull throughout the experiment (1-UP; LED Supply, Randolph, VT, USA). Cortical surface reflectance detected by OIS was collected with a lens (HR Plan Apo 0.5 × WD 136) through a 515LP emission filter on a Nikon SMZ 1500 stereomicroscope (Nikon Instruments, Melville, NY, USA). Images were acquired at 1–5 Hz using a high-sensitivity USB monochrome CCD camera (CCE-B013-U; Mightex, Pleasanton, CA, USA) with 4.65-micron square pixels and 1392 × 1040 pixel resolution.

Lateral to the thinned window two burr holes were drilled around the midpoint of the rectangle. These burr holes were deeper than the previously drilled skull region such that the dura was exposed but not broken. To record local field potentials (LFPs) an electrode (in a pulled glass pipette filled with saline) was inserted into one burr hole and attached to an amplifier. A separate ground wire, placed underneath the skin caudal to the skull, grounded this set up and LFPs were recorded for an hour to ensure a stable baseline and recovery from any surgically induced CSDs. After an hour of stabilization, a second pulled glass pipette was filled with 1 M KCl and placed into the more rostral burr hole, avoiding contact with the brain or the surrounding skull. An initial flow of KCl was pushed to begin and then an even flow was held so that there was a constant small pool of KCl that filled the burr hole. Excess liquid was removed with tissue paper applied next to the burr hole. Regardless of grouping the CSD recording continued for 3600 s after the initial drip of KCl. Mice were euthanized by anesthetic overdose followed by decapitation.

## Golgi staining

Golgi staining was performed according to the FD Rapid Golgi Stain kit (FD Neurotechnologies). For NTG or Veh-treated mice, they underwent the chronic NTG model and on day 10, 4, 24, and 48 hr after ACY-738 treatment or vehicle, mice were anesthetized with isoflurane, euthanized, brain/spinal cord was rapidly removed, and tissue was rinsed in ddH$_2$O. Tissue was then placed in the impregnation solution that was an equal amount of solutions A and B that was prepared at least 24 hr in

advance. After the first 24 hr the brain was placed in new impregnation solution and then stored for 1 week in the dark. The brains were then transferred to solution C, which was also replaced after the first 24 hr. After replacing solution C the brains were stored at room temperature for 72 hr more. Following solution C, brains were flash frozen in 2-methyl butane and cryostat cut at −20℃ into 100 μm slices. The slices were mounted onto gelatin coated slides and secured by a drop of solution C placed onto each slice. These slides were then left to dry naturally in the dark.

## Neurite tracing

After processing, images were taken at ×20 magnification and a Z-stack was created based on different levels of focal plane. After the Z-stack was created the FIJI program Simple Neurite Tracer was used to trace the processes of the neuron. While many of the neurons had some overlap with other analyzed neurons, Z-stacks of varying focus levels allowed for clearer tracing. A sample gif file of a Z-stack from a traced neuron is included and demonstrates how the change in focus allow for better determination of branching from overlapping neurons (*Animation 1*). Furthermore, after tracing the neurons were analyzed using Simple Neurite Tracer (*Longair et al., 2011*) software to assess the number of branch points from each neuron, overall length of the neuron, and Sholl Analysis. Sholl Analysis was performed by placing a center ROI point at the center of the soma and producing consecutive circles every 20 pixels/0.377 μm, for the entire body of the neuron. Intersections were counted based on the number of times a neurite crossed each of these consecutive circles. These data were compiled per neuron and then brought into one Masterfile. Male and female mice were used for a majority of studies, and no significant differences were observed in any of the key findings based on sex.

## Neuron Selection

Throughout tracing all tracers were blinded to which group the images belonged to. For all brain regions analyzed, six to eight relatively isolated neurons were randomly chosen per mouse. The selected neurons were fully impregnated with Golgi stain and relatively complete. An atlas was used along with clear anatomical markers to ensure the neurons were being taken from their described region of interest. Neurons characterized for the trigeminal nucleus caudalis region were taken only from the outer lamina of caudal sections. Neurons analyzed for somatosensory cortex were all taken from layer IV of the primary somatosensory barrel cortex. To ensure a homogenous cell population, only pyramidal cells were selected. The most complex neurons were chosen for analysis in all regions. Previously, it was shown that dendritic complexity was directly correlated to soma size. To ensure that the NTG group where not just smaller in size we directly compared soma diameter of neurons in the NTG and Veh group. There was no significant difference in soma size between these two groups (Veh 9.258 ±. 2515 and NTG 9.192 ±. 2782, student run t-test p=0.8608). Three individuals traced all cells. Interrater reliability was determined by having each tracer trace five neurons in their entirety. Pearson product correlations were accessed in three measures; number of branches, total dendritic length, and total intersection number through Sholl analysis and found to be 0.91, 0.94, and 0.95, respectively. All tracings of neurons were re-examined by the primary tracer (Z.B.) to assure quality control. Original neuronal traces can also be viewed at NeuroMorpho.org (http://neuromorpho.org/KeywordResult.jsp?count=837&keywords=%22bertels%22).

## Drug injections

**Animation 1.** Representational gif file of trigeminal nucleus caudalis (TNC) image at 20X magnification used to trace neurons. Images were taken through changes in z-stack focus and collapsed into a single image file. During tracing, different focal levels were used to better differentiate neurite components from one neuron to another.
https://elifesciences.org/articles/63076#video1

All injections were administered at 10 ml/kg volume, intraperitoneally (IP), unless otherwise indicated. ACY-738 was dissolved in a 5% DMSO saline solution, which was used as the vehicle control. RN-73 was dissolved in 10% DMSO, 10% Tween-80, and 80% saline and was injected 1 mg/kg or 10 mg/kg, this mixture was also used as the vehicle control group. ASV-85 was dissolved in 15% DMSO, 15% Tween-80, and then 70% saline, this mixture was also used as the vehicle control group. ASV-85 was injected at 1

mg/kg dose. TSA was dissolved in 20% DMSO solution in 80% 0.01M PBS and injected at a dose of 2 mg/kg, this same solution was used for the vehicle. Olcegepant was dissolved in saline solution and was injected at a 1 mg/kg dose. For the CSD experiments ACY-738 was injected 3 hr before starting the surgery so that it would reach its peak efficiency of 4 hr by the time the CSD event started.

## Quantitative RT-PCR

Total RNA was isolated from flash frozen brain punches using the RNeasy Plus Mini kit from Quiagen. RNA samples were reverse transcribed to single-stranded cDNA. cDNA transcription was used following the protocol from Superscript III (Life Technologies) and the TaqMan Gene Expression Assay system (Applied Biosystems). Glyceraldehyde-3-phosphate dehydrogenase (GAPDH, Hs02758991_g1) was used as a housekeeping gene. The threshold cycle (CT) of each target product was determined and CT values between HDAC6 transcripts and housekeeping genes were calculated ($\Delta$CT). The fold change ($2^{-\Delta\Delta CT}$) for each was calculated relative to the median $\Delta$CT from the saline control animals.

## Immunohistochemistry

Mice were anesthetized with Somnasol (100 µl/mouse; 390 mg/mL pentobarbital sodium; Henry Schein) and perfused intracardially with 15 ml of ice-cold phosphate-buffered saline (0.1 M PBS, pH 7.2) and subsequently 50 mL of ice-cold 4% paraformaldehyde (PFA) in 0.1M PBS (pH 7.4). Whole brain and trigeminal ganglia (TG) were harvested and overnight left to post-fix in 4% PFA/0.1M PBS at 4°C. Brain and TG were then cryoprotected in 30% sucrose in 0.1M PBS until they sunk. Brains were then flash frozen using 2-methyl butane over dry ice. Coronal sections of the trigeminal nucleus caudalis (TNC) and the somatosensory cortex were sliced on a cryostat at 20 µM and TG at 16 µM and immediately mounted onto slides. Slides were washed with PBST, then incubated with a blocking solution containing 5% normal donkey serum with PBST for 1 hr at room temperature. Slides were then incubated overnight at RT with the primary rabbit anti-HDAC6 antibody (1:500, courtesy of Tso-Pang Yao at Duke University) diluted in 1% NDSDT. Slides were subsequently washed with 1% NDST and then the secondary antibody was added for 2 hr at room temperature (donkey anti rabbit IgG, 1:2000). Slides were washed with 0.1 M phosphate buffer, and cover slipped with Mowiol-DAPI mounting medium. Images were taken by in a blinded manner using the EVOS FL Auto Cell Imaging system, using a ×40 objective.

## Western blots

Samples were taken from chronically treated NTG or Vehicle mice, which received an injection of ACY-738 or Vehicle on day 10. Samples were collected 4 hr post-ACY/VEH. TG, TNC, and SCtx was analyzed using traditional western blot analysis while Nac and spinal cord samples were analyzed at a later time using the ProteinSimple Wes. Spinal cord samples were a combination of cervical and lumbar spinal cord sections. Protein concentrations were assessed using a Nanodrop 2000c spectrophotometer and equal quantities were loaded onto each Stain-Free acrylamide gel for SDS-PAGE (Bio-Rad, Hercules, CA, USA). The gels were subsequently transferred to nitrocellulose membranes (Bio-Rad, Hercules, CA USA) for western blotting. The membranes were blocked with 5% non-fat dry milk diluted in TBS-T (10 mM Tris-HCl, 159 mM NaCl, and 0.1% Tween 20, pH 7.4) for 1 hr. Following the blocking step, membranes were washed with Tris-buffered saline/Tween 20 and then incubated with an anti-acetyl-$\alpha$-tubulin antibody (Lysine-40) (Sigma Clone 6-11B1), $\alpha$-tubulin (Sigma), overnight at 4°C. Membranes were washed with TBS-T and incubated with a secondary antibody [HRP-linked anti-mouse antibody IgG F(ab')two or HRP-linked anti-rabbit antibody IgG F(ab')2] (Jackson ImmunoResearch, West Grove, PA, USA, catalog #115-036-072 for mouse, and catalog #111-036-047 for rabbit,) for 1 hr at room temperature, washed, and developed using ECL Luminata Forte chemiluminescent reagent (Millipore, Billerica, MA, USA). Blots were imaged using a Chemidoc computerized densitometer (Bio-Rad, Hercules, CA, USA) and quantified by ImageLab 3.0 software (Bio-Rad, Hercules, CA, USA). In all experiments, the original gels were visualized using BioRad stainfree technology to verify protein loading. For the spinal cord and nucleus accumbens, samples were prepared and run on the ProteinSimple Instruments Wes System according to the manufacturer's instructions. Images for these samples were also measured and visualized using the same system.

## Statistical analysis

Sample size was calculated by power analysis and previous experience. Since we investigated changes at the cellular level, an individual neuron represented a single sample (*Espallergues et al., 2012*; *Moonat et al., 2013*). Data analysis was performed using GraphPad Prism version 8.00 (GraphPad, San Diego, CA). The level of significance (α) for all tests was set to 0.05. Post hoc analysis was conducted using Holm-Sidak post hoc test to correct for multiple comparisons. Post hoc analysis was only performed when F values achieved p<0.05. All values in the text are reported as mean ± SEM. Detailed statistical analysis can be found in *Supplementary file 2*.

## Acknowledgements

This work was funded by NIH grants: NS109862 (AAP), DA040688 (AAP), AT009169 (MMR), VA grant VA BX00149 (MMR), an Amgen Competitive Migraine Grant (AAP), UICentre for Drug Discovery (AAP, PP). MMR is a VA Research Career Scientist. ZB and EM are members of the UIC Graduate Program in Neuroscience. We thank Tso-Pang Yao from Duke University for the HDAC6-selective antibody.

## Additional information

### Funding

| Funder | Grant reference number | Author |
|---|---|---|
| National Institute of Neurological Disorders and Stroke | NS109862 | Amynah A Pradhan |
| National Institute on Drug Abuse | DA040688 | Amynah A Pradhan |
| National Center for Complementary and Integrative Health | AT009169 | Mark M Rasenick |
| Center for Integrated Healthcare, U.S. Department of Veterans Affairs | BX00149 | Mark M Rasenick |
| Amgen Foundation | | Amynah A Pradhan |
| Center for Clinical and Translational Science, University of Illinois at Chicago | | Amynah A Pradhan |

The funders had no role in study design, data collection and interpretation, or the decision to submit the work for publication.

### Author contributions

Zachariah Bertels, Conceptualization, Data curation, Formal analysis, Validation, Investigation, Methodology, Writing - original draft, Project administration; Harinder Singh, Conceptualization, Data curation, Formal analysis, Investigation; Isaac Dripps, Kendra Siegersma, Catherine Conway, Data curation, Formal analysis, Validation, Investigation, Methodology; Alycia F Tipton, Zoie Sheets, Data curation, Formal analysis, Investigation, Methodology; Wiktor D Witkowski, Validation, Investigation, Methodology; Pal Shah, Investigation, Methodology; Elizaveta Mangutov, Mei Ao, Data curation, Investigation; Valentina Petukhova, Resources, Data curation; Bhargava Karumudi, Resources, Data curation, Validation; Pavel A Petukhov, Conceptualization, Resources, Data curation, Writing - review and editing; Serapio M Baca, Conceptualization, Data curation, Formal analysis, Methodology, Project administration, Writing - review and editing; Mark M Rasenick, Conceptualization, Resources, Supervision, Funding acquisition, Methodology, Project administration, Writing - review and editing; Amynah A Pradhan, Conceptualization, Data curation, Formal analysis, Supervision, Funding acquisition, Investigation, Methodology, Writing - original draft, Project administration, Writing - review and editing

## Author ORCIDs
Harinder Singh (iD) http://orcid.org/0000-0002-0160-1575
Amynah A Pradhan (iD) https://orcid.org/0000-0001-9691-2976

## Ethics
Animal experimentation: This study was performed in strict accordance with the recommendations in the Guide for the Care and Use of Laboratory Animals of the National Institutes of Health. All of the animals were handled according to approved institutional animal care and use committee (IACUC) protocols (#18-250) of the University of Illinois at Chicago.

## Decision letter and Author response
Decision letter https://doi.org/10.7554/eLife.63076.sa1
Author response https://doi.org/10.7554/eLife.63076.sa2

## Additional files

### Supplementary files
• Supplementary file 1. Summary of HDAC inhibition by ASV-85.

• Supplementary file 2. Statistical analysis.

• Transparent reporting form

## Data availability
All data generated or analysed during this study are included in the manuscript and supporting files.

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
