## [Decision Letter]

**Acceptance summary:**

The authors demonstrate first describe profound changes in neuronal branching in two different preclinical models of migraine, nitroglycerine- and spreading depression-induced. Remarkably, with inhibitors of HDAC6, they next show significant improvement in this phenotype, as well as reduction of the mechanical hypersensitivity that occurs, which together raise the possibility that HDAC6 inhibitors are a potential novel therapeutic for migraine.

**Decision letter after peer review:**

Thank you for submitting your article "Neuronal complexity is attenuated in chronic migraine and restored by HDAC6 inhibition" for consideration by *eLife*. Your article has been reviewed by 3 peer reviewers, and the evaluation has been overseen by a Reviewing Editor and Kate Wassum as the Senior Editor. The following individual involved in review of your submission has agreed to reveal their identity: Maggie Waung (Reviewer #1).

The reviewers have discussed the reviews with one another and the Reviewing Editor has drafted this decision to help you prepare a revised submission.

There is considerable enthusiasm for your manuscript, in particular, the important findings implicating HDAC inhibition of migraine associated structural changes in the brain and its effects on behavior. However, there is strong agreement among the Reviewers that more experimental data are needed to establish a tighter connection between neuronal complexity and changes in allodynia/CSD. The manuscript clearly presents separate features: neuronal complexity based on histology; HDAC6 elevation/inhibition; behavioral endpoints. However, it does not adequately establish cause and effect relationships among these components. Additional experiments that strengthen the correlation are essential to a successful revision. Specific experimental questions are detailed below.

Title: One Reviewer suggests that the title of the manuscript should more specifically indicate preclinical migraine models as opposed to the current description as chronic migraine.

Specific comments from the reviewers include:

The efficacy of ACY for behavior is shown at 4 and 24 hours, but histology of neuronal architecture is only shown at 4 hours. Does ACY-738-induced increase in neurite complexity last beyond 4 hours? Is this rapid efficacy consistent with the proposed mechanism? In other words, is it consistent for HDAC6 inhibition to normalize structural changes in 4 hours when those changes developed over 9 days?

As the effects of ACY in behavior are no longer present at 48 hours, one would predict that the decreased neuronal complexity returns at 48 hours, and is correlated with loss of efficacy in behavior. However these data are not present. This is an important missing piece of data to evaluate how closely the changes in neuronal architecture explain the behavioral effects.

Experiments providing longitudinal data about NTG and ACY's effects on neuronal complexity or a demonstration that targeted inhibition of HDAC6 during NTG administration prevents chronic allodynia would greatly strengthen the connection.

Please clarify how and to what extent blinding was achieved (e.g. with respect to testing for allodynia and drug administration).

Comparison representative Golgi stains from different treatment conditions should be provided, including sample 20X images. Based on the one image provided, neurons traced appear overlapping, which may affect tracing results.

The lack of change in the dorsal horn suggests that NTG selectively activates the trigeminal pain system, but there is no mechanism by which this could easily be explained. Additionally, not all noxious input from the head is signaled through the trigeminal, some is signaled through upper cervical DRG. At the very least it would be informative to know whether the observed findings occur in upper cervical DRG or whether they are entirely restricted to the trigeminal system.

Related to the above question, does ACY increase acetylated α-tubulin in the NAc and spinal cord, or are these effects specific to the TG, TNC, and cortex?

Sample western blots for Supplementary Figure 2 should be provided.

For the Discussion:

Several figures show that ACY increases neuronal complexity in sham animals. This suggests that ACY has an overall trophic effect on the nervous system. If true, this would mean that ACY does not specifically counteract the effects of NTG and CSD on neuronal complexity, but reverses these changes due to a general trophic effect across the nervous system. This may be consistent with the lack of increase in HDAC6 outside the TG in the NTG model. In other words, there is no requirement that HDAC6 be elevated in a specific region for neuronal complexity to decrease, because HDAC6 inhibition will increase neuronal complexity globally, regardless of levels of expression. This requires more discussion as it changes the interpretation and framing of the findings.

You report that a single olcegepant treatment after 9 days of NTG attenuates behavior and normalizes structural changes within 4 hours. ACY and anti-CGRP are clearly 2 mechanistically distinct approaches achieving the same endpoints, at the same times, suggesting that there is no need to inhibit HDAC6 directly to achieve these results.

Are the half-lives of the alternate HDAC inhibitors known? ACY has a 12 min half life and causes 24 hours of behavioral efficacy, but the other HDAC inhibitors effects were shorter lasting. It seems unlikely that their half-lives are shorter than 12 min, as this is already fairly rapid metabolism.

Since acetylated tubulin is important for touch, and KO of a-TAT1 leads to profound deficits in touch, how do you know that there is relief of cephalic pain with HDAC6 inhibition vs. loss of touch sensation, thus affecting assays of mechanical allodynia?

Consider a discussion about how an intervention that increases neurite growth might decrease CSD frequency. It also appears that CSD duration is increased with HDAC6 inhibition.

Similarly, how does a reduction in TNC neurites explain the physiological increase in TNC activity seen in other rodent models of headache (and in humans)?

Both males and female mice were used- is there a sex difference in the results?

Please ensure full statistical reporting within the manuscript, e.g., t values, f values, degrees of freedom, p values etc.

---

## [Author Response]

There is considerable enthusiasm for your manuscript, in particular, the important findings implicating HDAC inhibition of migraine associated structural changes in the brain and its effects on behavior. However, there is strong agreement among the Reviewers that more experimental data are needed to establish a tighter connection between neuronal complexity and changes in allodynia/CSD. The manuscript clearly presents separate features: neuronal complexity based on histology; HDAC6 elevation/inhibition; behavioral endpoints. However, it does not adequately establish cause and effect relationships among these components. Additional experiments that strengthen the correlation are essential to a successful revision. Specific experimental questions are detailed below.Title: One Reviewer suggests that the title of the manuscript should more specifically indicate preclinical migraine models as opposed to the current description as chronic migraine.

Agreed. We have changed the title to “Neuronal complexity is attenuated in preclinical models of migraine and restored by HDAC6 inhibition”.

Specific comments from the reviewers include:The efficacy of ACY for behavior is shown at 4 and 24 hours, but histology of neuronal architecture is only shown at 4 hours. Does ACY-738-induced increase in neurite complexity last beyond 4 hours? Is this rapid efficacy consistent with the proposed mechanism? In other words, is it consistent for HDAC6 inhibition to normalize structural changes in 4 hours when those changes developed over 9 days?As the effects of ACY in behavior are no longer present at 48 hours, one would predict that the decreased neuronal complexity returns at 48 hours, and is correlated with loss of efficacy in behavior. However these data are not present. This is an important missing piece of data to evaluate how closely the changes in neuronal architecture explain the behavioral effects.

We agree that experiments addressing the time course of ACY-738 strengthen our findings. We have performed two additional experiments in which mice are treated chronically with NTG/VEH for 9 days. On day 10 mice were injected with ACY-738 or vehicle and subsequently sacrificed 24 or 48 hrs post-treatment and the trigeminal nucleus caudalis was analyzed using Golgi staining. We find that the timing of the anti-allodynic effects of ACY-738 corresponds with its normalizing effect on neuronal cytoarchitecture. ACY-738 inhibits migraine-associated pain at 4 and 24h post-injection, and this effect is lost 48h post-injection. In terms of Golgi staining: at 4h, both vehicle-ACY and NTG-ACY groups show increased neuronal complexity (Figure 2), while at 24h only the NTG-ACY group has increased neuronal complexity (Figure 3C-E); and at 48h the NTG-ACY group is similar to the NTG-VEH group (ie decreased neuronal complexity relative to VEH-VEH controls, Figure 3F-H).

Experiments providing longitudinal data about NTG and ACY's effects on neuronal complexity or a demonstration that targeted inhibition of HDAC6 during NTG administration prevents chronic allodynia would greatly strengthen the connection.

To address this, we performed an additional experiment in which ACY-738 was administered following each NTG injection in the chronic migraine paradigm. In this case, chronic treatment with ACY-738 prevented the development of NTG-induced allodynia (NTG-VEH vs NTG-ACY738, Figure 4D). These data indicate that sustained HDAC6 inhibition can prevent the development of chronic-migraine associated pain.

Please clarify how and to what extent blinding was achieved (e.g. with respect to testing for allodynia and drug administration).

We have updated this in the methods. While the same experimenter administered both the injections and performed the mechanical threshold testing, these were done at different time points and the data kept separately. Thus, the experimenter was essentially blinded at the time of testing.

Comparison representative Golgi stains from different treatment conditions should be provided, including sample 20X images. Based on the one image provided, neurons traced appear overlapping, which may affect tracing results.

Gif files showing how the z stack focus was changed to allow for tracing were added to the manuscript (Animation 1). These stack images allow for accurate tracing even in the cases where there are some partial overlap of neurons.

The lack of change in the dorsal horn suggests that NTG selectively activates the trigeminal pain system, but there is no mechanism by which this could easily be explained. Additionally, not all noxious input from the head is signaled through the trigeminal, some is signaled through upper cervical DRG. At the very least it would be informative to know whether the observed findings occur in upper cervical DRG or whether they are entirely restricted to the trigeminal system.

Agreed. We could not use Golgi stain on ganglia neurons (TG or dorsal root ganglia), however, we did examine the cervical spinal cord. We determined if chronic NTG resulted in altered neuronal cytoarchitecture in the dorsal horn of C1-C3. Similar to the lumbar spinal cord we did not observe any significant difference between vehicle and NTG treated groups (Figure 1—figure supplement 1). These results suggest that in this model altered cytoarchitectural dynamics are more prominent in the brain and not necessarily throughout the whole pain circuit. Future studies will investigate how neuronal plasticity is altered in some regions, but not others.

Related to the above question, does ACY increase acetylated α-tubulin in the NAc and spinal cord, or are these effects specific to the TG, TNC, and cortex?

Additional experiments were added to examine the effect of ACY-738 on tubulin in both the nucleus accumbens and the spinal cord. In both regions we saw an increase in acetylated tubulin following administration of ACY-738. This indicates that the effects of increased acetylated tubulin are not specific to migraine specific areas. Please see Figure 2—figure supplement 1.

Sample western blots for Supplementary Figure 2 should be provided.

We have adjusted this figure, now called Figure 2—figure supplement 1, accordingly.

For the Discussion:Several figures show that ACY increases neuronal complexity in sham animals. This suggests that ACY has an overall trophic effect on the nervous system. If true, this would mean that ACY does not specifically counteract the effects of NTG and CSD on neuronal complexity, but reverses these changes due to a general trophic effect across the nervous system. This may be consistent with the lack of increase in HDAC6 outside the TG in the NTG model. In other words, there is no requirement that HDAC6 be elevated in a specific region for neuronal complexity to decrease, because HDAC6 inhibition will increase neuronal complexity globally, regardless of levels of expression. This requires more discussion as it changes the interpretation and framing of the findings.

Although HDAC6 inhibitor does increase neuronal complexity in vehicle treated groups, this increased trophism does not result in any behavioral consequence in the endpoints measured. In addition, the time course that we have added to this manuscript show that this effect in the vehicle group is short lasting (4h) relative to ACY treatment in the NTG group (24h). We have updated the discussion to reflect the point that HDAC6 inhibition might be one of many ways in which blunted neuronal complexity associated with chronic migraine can be corrected. Importantly, as HDAC6 directly affects tubulin dynamics, HDAC6 inhibitors represent a more targeted strategy to restore neuronal complexity. Our findings open the possibility of examining HDAC6 and other determinants of cytoarchitectural dynamics as potential migraine therapies.

You report that a single olcegepant treatment after 9 days of NTG attenuates behavior and normalizes structural changes within 4 hours. ACY and anti-CGRP are clearly 2 mechanistically distinct approaches achieving the same endpoints, at the same times, suggesting that there is no need to inhibit HDAC6 directly to achieve these results.

Agreed. Our data suggest that blunted neuronal complexity could be a characteristic of chronic migraine, and restoration of this complexity a sign of effective therapy. As mentioned in the above comment there appear to be multiple ways to restore neuronal complexity, including HDAC6 inhibition and CGRP antagonists. We have rewritten the Discussion to reflect this point, and future studies will focus on common processes/pathways between these mechanistically distinct targets.

Are the half-lives of the alternate HDAC inhibitors known? ACY has a 12 min half life and causes 24 hours of behavioral efficacy, but the other HDAC inhibitors effects were shorter lasting. It seems unlikely that their half-lives are shorter than 12 min, as this is already fairly rapid metabolism.

Preliminary measurement of both RN73 and ASV85 show that it is still present in the plasma and to a lesser extent the brain at 40 minutes post-injection. Only the 20 and 40 min time points have been measured and peak levels are seen at the earlier time point. ASV85 has a slightly better brain penetrance. We are happy to include this data in the Materials and methods, but it is preliminary.

Since acetylated tubulin is important for touch, and KO of a-TAT1 leads to profound deficits in touch, how do you know that there is relief of cephalic pain with HDAC6 inhibition vs. loss of touch sensation, thus affecting assays of mechanical allodynia?

The reviewer brings up a good point, which we have now addressed in the discussion. In mice chronically treated with vehicle (control for NTG), acute ACY-738 does not affect mechanical thresholds at 4, 24, or 72h (Figure 3A). In addition, even when ACY-738 is given chronically we do not observe any change in the ACY-VEH group, and the NTG-ACY group maintains normal mechanical responses (Figure 4D), while a loss of sensation may result in a seemingly antinociceptive effect. Furthermore, three other groups have used the same and different HDAC6 inhibitors and none report changes in responding in control groups.

Consider a discussion about how an intervention that increases neurite growth might decrease CSD frequency. It also appears that CSD duration is increased with HDAC6 inhibition.

Agreed. We have added to the discussion on how HDAC6 inhibition may disrupt the wave path of CSD events.

Similarly, how does a reduction in TNC neurites explain the physiological increase in TNC activity seen in other rodent models of headache (and in humans)?

In the discussion we outline a few possible mechanisms for how reduced neuronal complexity could still result in increased activity in these head pain-processing regions. For example, decreases neuronal complexity would not only weaken or prevent some neuronal connections/synapses, the decreased flexibility may also strengthen other synapses as the neuron might be in a more fixed state. The strengthening of these synapses would still be measured as increased neuronal activity.

Both males and female mice were used- is there a sex difference in the results?

We analyzed our data carefully for sex differences but did not find anything significant.

Please ensure full statistical reporting within the manuscript, e.g., t values, f values, degrees of freedom, p values etc.

Greater specificity and statistical reporting was added to the figure legends to better convey our findings.